# FIP200 restricts RNA virus infection by facilitating RIG-I activation

Lingyan Wang[1,5], Kun Song[1,5], Wenzhuo Hao[1,5], Yakun Wu[1,5], Girish Patil[2], Fang Hua[1], Yiwen Sun[1], Chaoqun Huang[2], Jerry Ritchey[3], Clinton Jones[3], Lin Liu[2], Jun-Lin Guan[4] & Shitao Li[1✉]

Retinoic acid-inducible gene I (RIG-I) senses viral RNA and instigates an innate immune signaling cascade to induce type I interferon expression. Currently, the regulatory mechanisms controlling RIG-I activation remain to be fully elucidated. Here we show that the FAK family kinase-interacting protein of 200 kDa (FIP200) facilitates RIG-I activation. FIP200 deficiency impaired RIG-I signaling and increased host susceptibility to RNA virus infection. In vivo studies further demonstrated FIP200 knockout mice were more susceptible to RNA virus infection due to the reduced innate immune response. Mechanistic studies revealed that FIP200 competed with the helicase domain of RIG-I for interaction with the two tandem caspase activation and recruitment domains (2CARD), thereby facilitating the release of 2CARD from the suppression status. Furthermore, FIP200 formed a dimer and facilitated 2CARD oligomerization, thereby promoting RIG-I activation. Taken together, our study defines FIP200 as an innate immune signaling molecule that positively regulates RIG-I activation.

[1] Department of Microbiology and Immunology, Tulane University, New Orleans, LA, USA. [2] Department of Physiological Sciences, Oklahoma State University, Stillwater, OK, USA. [3] Department of Pathology, Oklahoma State University, Stillwater, OK, USA. [4] Department of Cancer Biology, University of Cincinnati College of Medicine, Cincinnati, OH, USA. [5] These authors contributed equally: Lingyan Wang, Kun Song, Wenzhuo Hao, Yakun Wu. ✉email: sli38@tulane.edu

The innate immune system uses pattern recognition receptors in different cellular compartments to sense microbial components that mark invading viruses[1]. The RIG-I-like receptors are a small family of cytosolic RNA sensors, including RIG-I, MDA5, and LGP2. RIG-I recognizes double-stranded RNA (dsRNA) or 5′ triphosphate RNA in the cytoplasm generated by RNA viruses. The engagement of viral RNA induces the conformational change of RIG-I and several post-translational modifications, which leads to the oligomerization of RIG-I[2,3]. The oligomerized RIG-I translocates to the mitochondria and binds the mitochondrial antiviral signaling protein (MAVS). The binding results in the oligomerization of MAVS and the recruitment of TANK-binding kinase 1 (TBK1). Subsequently, TBK1 proteins oligomerize and trans-phosphorylate each other for activation. Activated TBK1 further phosphorylates interferon regulatory factors (IRFs). Phosphorylated IRFs translocate to the nucleus and form active transcriptional complexes to activate type I interferon (IFN) expression[1].

RIG-I consists of the N-terminal 2CARD, followed by a DExD-box RNA helicase domain and a carboxy-terminal domain (CTD). The interaction and post-translational modifications of these domains regulate RIG-I activation. In the absence of RNA ligand, RIG-I is auto-repressed because the 2CARD is prevented from forming the active conformation due to the binding of the helicase domain. Recent studies showed that RIG-I is constitutively phosphorylated at the 2CARD and CTD domains, which further prevents RIG-I from activation[4–6]. Once the CTD captures dsRNA, the helicase domain alters its conformation to a closed form with a high affinity for dsRNA. Concomitantly the 2CARD is released. Subsequently, the 2CARD is dephosphorylated by protein phosphatase 1 alpha (PP1α) and PP1γ, which results in a rearrangement of the 2CARD[7]. The 2CARD and CTD domain are then subject to K63-linked polyubiquitintion by several ubiquitin E3 ligases, including TRIM25[8], MEX3C[9], RIPLET[10], and TRIM4[11]. The free or conjugated K63-linked polyubiquitin induces CARD tetramers and facilitates RIG-I multimerization and filamentation, thereby activating RIG-I[12–14]. Nonetheless, the regulatory mechanisms controlling RIG-I activation remain to be fully elucidated.

FIP200, also known as the retinoblastoma 1-inducible coiled coil 1, is an evolutionarily conserved protein consisting of an N-terminal region, a large coiled-coil domain (CC) in the middle, and a C-terminal ATG11-like domain (ATG)[15]. FIP200 interacts with multiple intracellular signaling proteins[15], including FAK, ATG13, and TRAF2. By interaction with ATG13, FIP200 forms a complex with UNC-51-like kinase 1 (ULK1), which initiates autophagosome nucleation[16,17]. In addition to the autophagic role, FIP200 participates in several other physiological and pathological processes, such as cell migration, proliferation, tumorigenesis, and apoptosis. A recent study showed that FIP200 limited picornavirus replication, and its antiviral activity was independent of autophagy, suggesting a link to host defense[18]. However, the mechanism by which FIP200 regulates host defense is unknown.

In this study, we show that FIP200 is essential for RIG-I activation and host defense to RNA virus infection. We first identified RIG-I as an interactor of FIP200 by proteomics. We then determined the domains required for FIP200-RIG-I interaction and found the interaction was mediated by the CARD of RIG-I and the Claw domain of FIP200. Next, we used FIP200 knockout cells and knockout mice to examine the role of FIP200 in RIG-I signaling pathway. Deficiency of FIP200 impaired dsRNA-induced type I IFN expression and increased host susceptibility to vesicular stomatitis virus (VSV) and human coronavirus (HCoV) OC43 infection. Further mechanistic analyses found that FIP200 potentiated RIG-I de-repression by competing with the helicase-CTD domain of RIG-I for CARD binding. FIP200 also dimerized and facilitated the dimerization of 2CARD.

## Results

**FIP200 interacts and co-localizes with RIG-I**. To discover FIP200-interacting proteins, we performed the affinity purification coupled with mass spectrometry (AP-MS) analysis of the FIP200 protein complex (Supplementary Fig. 1a). First, FLAG-tagged FIP200 was transfected into HEK293 cells to generate a stable cell line. After the stable cell line was established, FIP200 protein complexes were purified by affinity purification and then were analyzed by mass spectrometry. The AP-MS was biologically repeated twice. To efficiently reduce false positives in AP-MS, we adopted the well-established statistical method SAINT[19]. Using a stringent statistical SAINT score cutoff of 0.89 ($P < 0.01$), we identified twelve high-confidence candidate interacting proteins, including two well-known interactors, ATG13 and ATG101[20–22] (Supplementary Fig. 1b; Supplementary Data 1), which substantiates the high quality of our FIP200 protein interaction network.

Unexpectedly, the proteomics also found that RIG-I interacted with FIP200 (Supplementary Fig. 1b). The interaction between RIG-I and FIP200 was validated by co-immunoprecipitation (co-IP) in HEK293 cells (Supplementary Fig. 1c). We also examined the endogenous protein interaction between RIG-I and FIP200 in RAW 264.7 macrophages stimulated with poly(I:C), an RNA ligand for RIG-I. As shown in Fig. 1a, the interaction between endogenous RIG-I and FIP200 was enhanced after stimulation, suggesting that poly(I:C) induces FIP200-RIG-I interaction. Next, we examined the subcellular localization of FLAG-tagged FIP200 and HA-tagged RIG-I in A549 lung epithelial cells. Immunofluorescence assay (IFA) displayed a cytosolic co-localization between FIP200 and RIG-I (Supplementary Fig. 1d). Consistently, endogenous FIP200 and RIG-I also co-localized in the cytoplasm of A549 cells (Fig. 1b). However, IFA cannot determine their interaction in situ, as both FIP200 and RIG-I are diffusely expressed in the cytoplasm. Therefore, we further performed proximity ligation assay (PLA) to determine the in situ interaction. The PLA assay showed that poly(I:C) promoted the in situ interaction between endogenous FIP200 and RIG-I and increased their interactions at the mitochondria (Fig. 1c and Supplementary Fig. 1e). Overall, these data demonstrate that RIG-I interacts with FIP200 upon ligand stimulation.

**FIP200 interacts with the CARD domain of RIG-I**. To determine the domain responsible for the interaction, we first examined the interaction between FIP200 and the domains of RIG-I (Fig. 1d). Co-IP found that the 2CARD was sufficient for the interaction with FIP200 (Fig. 1e). As the 2CARD is conserved in MDA5, the 2CARD from MDA5 also interacted with RIG-I (Supplementary Fig. 1f). Similarly, we determined the domain of FIP200 required for RIG-I interaction (Fig. 1f). Deletion of the C-terminal ATG domain dramatically impaired the interaction with RIG-I (Fig. 1g), suggesting that the ATG domain is required for RIG-I interaction. Next, we co-transfected FIP200 or the ATG domain with the 2CARD of RIG-I into HEK293 cells. Co-IP found the ATG domain alone was sufficient for the interaction with 2CARD (Supplementary Fig. 1g). To further determine whether this interaction is direct, we purified GST-tagged 2CARD and His-tagged ATG from *E. coli*. In vitro pull-down studies confirmed the direct interaction between 2CARD and ATG (Supplementary Fig. 1h). Since the 2CARD has two CARD domains, we further determined which CARD was responsible for the interaction. A similar in vitro pull-down was performed between His-tagged ATG and MBP-tagged CARD1 or CARD2. As shown in Fig. 1h, both CARD1 and CARD2 interacted with the ATG domain. A recent structural study showed that the ATG

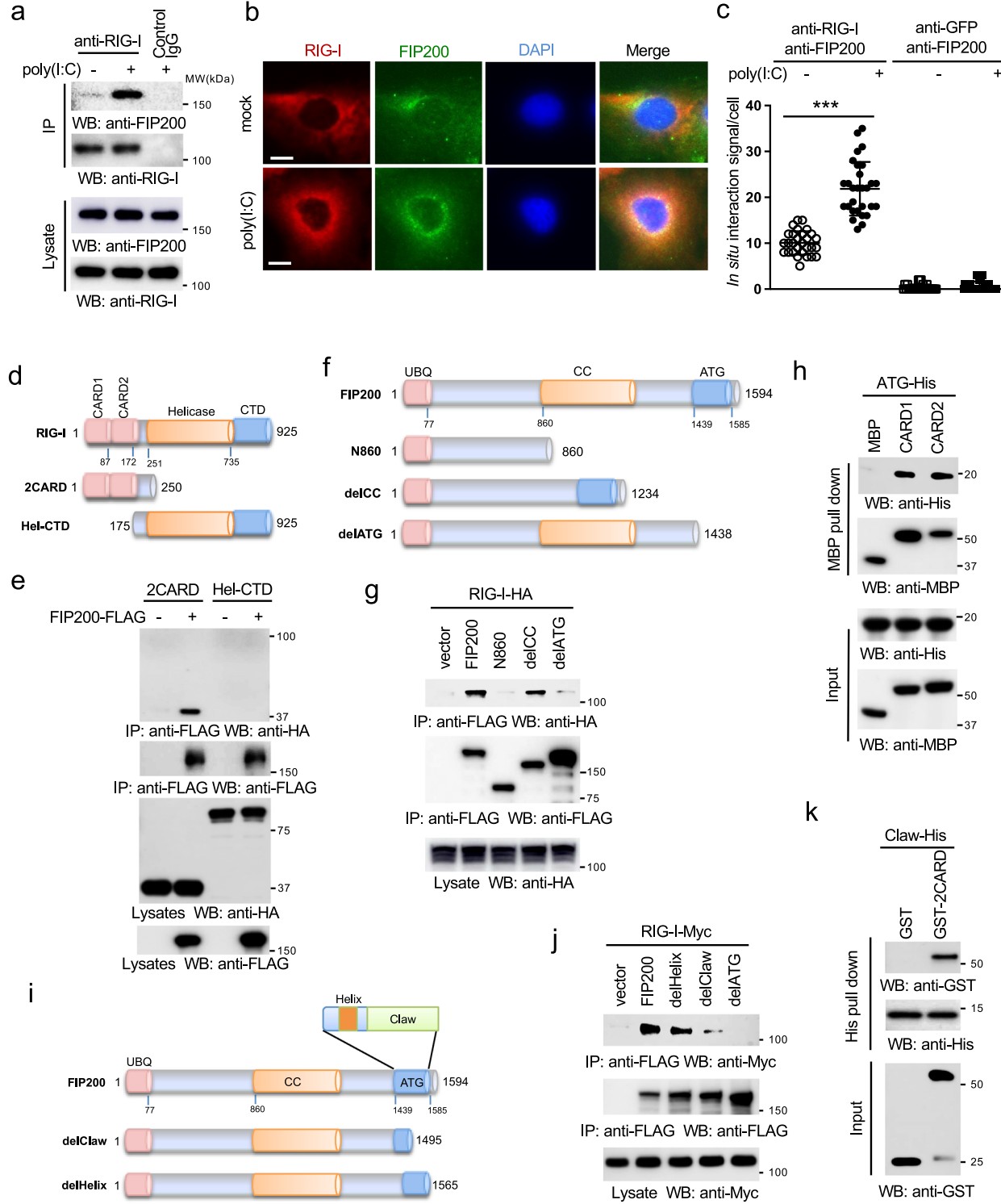

domain contains two smaller distinct domains, an alpha-helix and a claw shape structure (Claw) (Fig. 1i)[23]. Deletion of Claw, but not helix, reduced the binding of FIP200 to RIG-I (Fig. 1j). In vitro pull-down further found the direct interaction between the 2CARD and the Claw domain (Fig. 1k). The combined data suggest that Claw domain and CARD domain mediate the interaction between FIP200 and RIG-I.

**Ectopic expression of FIP200 promotes RIG-I activation**. The interaction between FIP200 and RIG-I suggests a potential link of

FIP200 to the RIG-I signaling pathway. To test this hypothesis, we first examined the effects of ectopic expression of FIP200 on RIG-I-induced NF-κB and interferon reporter activities. RIG-I was co-transfected with FIP200 together with the NF-κB reporter or the interferon-stimulated response element (ISRE) reporter into HEK293 cells. A low amount of RIG-I was used to induce minimal reporter activity. FIP200 synergistically induced both NF-κB and ISRE reporter activities with RIG-I (Fig. 2a). Next, we examined the effects of FIP200 on mRNA expression of IFNβ and two interferon-stimulated genes (ISGs), IP10 and RANTES. Consistently, FIP200 synergistically induced the mRNA

**Fig. 1 FIP200 interacts and co-localizes with RIG-I. a** RAW 264.7 macrophages were transfected with 1 μg ml$^{-1}$ poly(I:C). Two hours later, cells were harvested and immunoprecipitated with control IgG or the anti-RIG-I antibody. The IP and lysate samples were blotted with the indicated antibodies. Molecular weights (MW) are indicated. **b** A549 cells were transfected with 1 μg ml$^{-1}$ poly(I:C). Two hours later, cells were fixed and stained with the anti-FIP200 antibody (green), anti-RIG-I antibody (red), and DAPI (blue). Bar = 10 μm. **c** A549 cells were transfected with 1 μg ml$^{-1}$ poly(I:C). Two hours later, cells were fixed, and the proximity ligation assays were performed. ***$P < 0.001$, by one-way ANOVA, followed by Tukey's multiple comparison test. **d** Schematics of the RIG-I mutants. CARD: caspase activation and recruitment domain; CTD: C-terminal domain; Hel-CTD: helicase domain plus CTD. **e** FLAG-tagged FIP200 (FIP200-FLAG) was transfected with the indicated HA-tagged RIG-I mutants into HEK293 cells. After 48 h, cell lysates were collected, and then immunoprecipitated and blotted as indicated. **f** Schematics of the FIP200 mutants. UBQ: ubiquitin-like; CC: coiled coil; ATG: ATG11-like. **g** HA-tagged RIG-I (RIG-I-HA) was transfected with FLAG-tagged FIP200 and the indicated mutants into HEK293 cells. After 48 h, cell lysates were collected, and then immunoprecipitated and blotted as indicated. **h** His-tagged ATG domain (ATG-His) was mixed with MBP, MBP-tagged CARD1, or CARD2 in vitro at 4 °C for 16 h. Then, MBP pull-down assay was performed and blotted as indicated. All proteins were purified from *E. coli*. **i** Schematics of the alpha helix and Claw deletion mutant. **j** Myc-tagged RIG-I (RIG-I-Myc) was transfected with FLAG-tagged FIP200 and the indicated mutants into HEK293 cells. After 48 h, cell lysates were collected, and then immunoprecipitated and blotted as indicated. **k** Purified recombinant His-tagged Claw domain of FIP200 (Claw-His) was mixed with GST or GST-tagged 2CARD in vitro at 4 °C for 16 h. Then, His pull-down was performed and blotted as indicated.

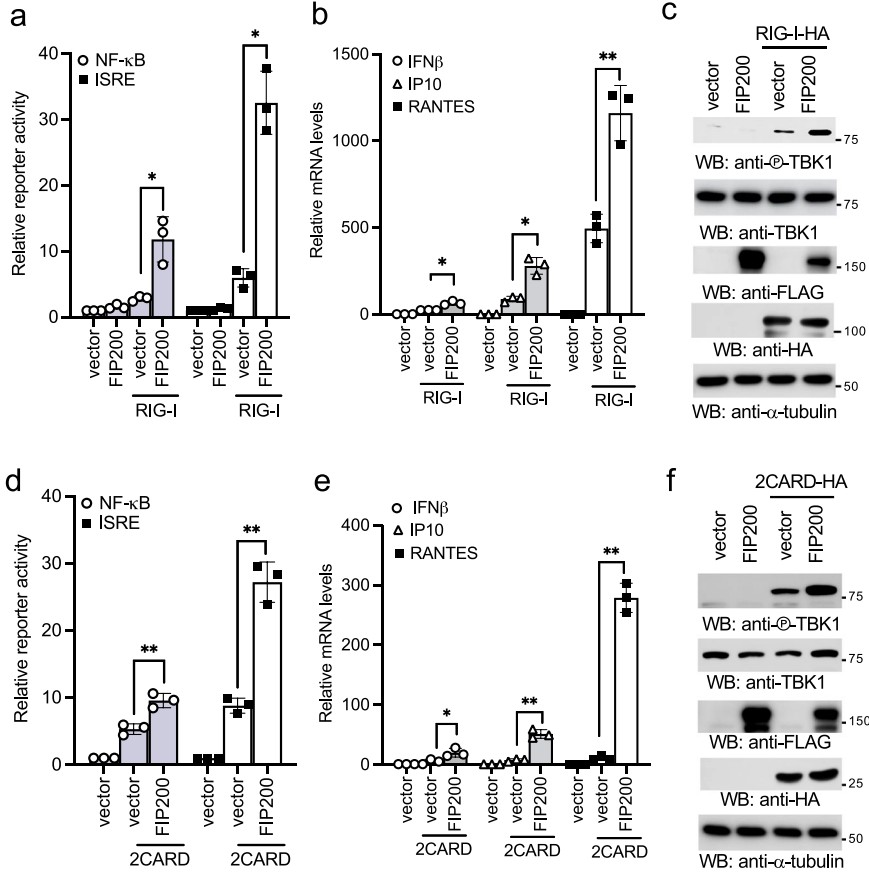

**Fig. 2 FIP200 activates type I IFN signaling synergistically with RIG-I. a** Two hundred ng of FLAG-tagged FIP200 was transfected with 20 ng of FLAG-tagged RIG-I and 20 ng of pRL-SV40 (*Renilla* luciferase as an internal control), together with 200 ng of pISRE-Luc or NF-κB-Luc into HEK293 cells. After 48 h, cells were collected, and the ratio of firefly luciferase to *Renilla* luciferase was calculated to determine the relative reporter activity. All experiments were biologically repeated three times. Data represent means ± s.d. of three independent experiments. (*$P < 0.05$, by paired two-tailed Student's *t* test). **b** RIG-I was transfected with vector or FIP200 into HEK293 cells. After 48 h, RNA was extracted, and real-time PCR for IFNβ, IP10, and RANTES was performed. All experiments were biologically repeated three times. Data represent means ± s.d. of three independent experiments. (*$P < 0.05$, **$P < 0.01$, by paired two-tailed Student's *t* test). **c** RIG-I was transfected with vector or FIP200 into HEK293 cells. After 48 h, cell lysates were blotted as indicated. **d** Two hundred ng of FLAG-tagged FIP200 was transfected with 20 ng of 2CARD-FLAG and 20 ng of pRL-SV40, together with 200 ng of pISRE-Luc or NF-κB-Luc into HEK293 cells. After 48 h, cells were collected, and the ratio of firefly luciferase to *Renilla* luciferase was calculated to determine the relative reporter activity. All experiments were biologically repeated three times. Data represent means ± s.d. of three independent experiments. (**$P < 0.01$, by paired two-tailed Student's *t* test). **e** The 2CARD was transfected with vector or FIP200 into HEK293 cells. After 48 h, RNA was extracted, and real-time PCR for IFNβ, IP10, and RANTES was performed. All experiments were biologically repeated three times. Data represent means ± s.d. of three independent experiments. (*$P < 0.05$, **$P < 0.01$, by paired two-tailed Student's *t* test). **f** The 2CARD was transfected with vector or FIP200 into HEK293 cells. After 48 h, cell lysates were blotted as indicated.

expression of IFNβ, IP10, and RANTES with RIG-I (Fig. 2b) and TBK1 phosphorylation (Fig. 2c). Since the 2CARD of RIG-I is constitutively active[24] and binds FIP200, we further examined the effects of FIP200 overexpression on 2CARD-mediated type I IFN response. Similarly, FIP200 and 2CARD synergistically induced both NF-κB and ISRE reporter activities (Fig. 2d), the mRNA expression of IFNβ, IP10, and RANTES (Fig. 2e), and TBK1 phosphorylation (Fig. 2f). Taken together, our data demonstrate that overexpression of FIP200 promotes RIG-I activation.

**The deficiency of FIP200 impairs RIG-I activation**. To corroborate the gain of function of FIP200, we examined the effects of FIP200 knockout on RIG-I signaling. First, we examined the protein levels of RIG-I, MAVS, TBK1, and IRF3 in FIP200 wild type vs. knockout mouse embryonic fibroblasts (MEFs). Western blotting showed comparable levels of these proteins in FIP200 wild type and knockout MEFs (Supplementary Fig. 2a), suggesting that FIP200 deficiency has little effect on the protein expression of the components of RIG-I signaling pathway. Furthermore, these cells showed comparable proliferation rate (Supplementary Fig. 2b). Next, we stimulated FIP200 wild type and knockout MEFs by transfection of low molecular weight poly (I:C) (hereinafter referred as poly(I:C)). The deficiency of FIP200 dramatically impaired the mRNA and protein expression of IFNα and IFNβ induced by poly(I:C) (Fig. 3a, Supplementary Fig. 2c–e). Similarly, another RIG-I ligand, 5′-ppp-dsRNA, failed to induce IFNβ expression in $Fip200^{-/-}$ MEFs (Fig. 3b). By contrast, FIP200 wild type and knockout MEFs showed comparable IFNβ responses to calf thymus DNA (ctDNA) and poly(dG:dC) (Fig. 3c, d), suggesting that FIP200 specifically regulates the cytosolic RNA sensing pathway. Consistently, Sendai virus (SeV) and the influenza A virus with NS1 deletion stimulated marginal IFNβ production in FIP200 knockout cells (Fig. 3e, f). Furthermore, FIP200 deficiency also impaired the mRNA expression of RANTES, IP10, and IRF7 (Fig. 3g, h, Supplementary Fig. 2f).

Next, we generated two FIP200 knockout cell lines in HEK293 cells by CRISPR, which proliferated at a comparable rate (Supplementary Fig. 2g) and exhibited similar protein levels of TBK1, RIG-I, MAVS, and IRF3 as in the parental HEK293 cells (Supplementary Fig. 2h). However, the poly(I:C)-induced mRNA expression of IFNβ as well as three ISGs, IP10, OASL, and IFIT1, was significantly reduced in FIP200 knockout cells (Fig. 3i, j, Supplementary Fig. 2i, j). Consistently, the IFNβ production induced by poly(I:C) and SeV was impaired in FIP200 knockout HEK293 cells (Fig. 3k, l). Furthermore, we examined poly(I:C)-induced TBK1 activation in FIP200 wild type vs. knockout cells. As shown in Fig. 3m, the serine 172 phosphorylation of TBK1 was almost abolished in FIP200 knockout cells.

Lastly, to examine the role of FIP200 in macrophages, we knocked out FIP200 in RAW264.7 cells by CRISPR. FIP200 deficiency has a minimal effect on the protein expression of TBK1, RIG-I, MAVS, and IRF3 in RAW264.7 cells (Supplementary Fig. 3a) and cell proliferation (Supplementary Fig. 3b). Next, we stimulated FIP200 wild type and knockout macrophages with different ligands. Consistent with the above observations in FIP200 knockout fibroblasts, FIP200 deficiency in macrophages also reduced IFNβ production induced by transfected low molecular weight poly(I:C), 5′-ppp-dsRNA and high molecular weight poly(I:C) (a MDA5 ligand) (Fig. 4a, b and Supplementary Fig. 3c), but had little effect on other ligands, including ctDNA, poly(dG:dC), DMXAA, LPS, and poly(A:U) (a TLR3 ligand) (Fig. 4c–f and Supplementary Fig. 3d). Furthermore, FIP200 knockout macrophages produced much less IFNβ after the infection of SeV and IAV, but not the d109 mutant of human herpes simplex virus type 1 (HSV-1) in

which the IFN-suppression viral genes are deleted[25] (Fig. 4g, h, Supplementary Fig. 3e). RNA sequencing analysis found that FIP200 deficiency also impaired the mRNA expression of type I interferons and ISGs (Fig. 4i), such as IFNβ, RANTES and IP10, which were validated by real-time PCR (Fig. 4j, k, l). Taken together, these data suggest that FIP200 is essential for RIG-I-mediated innate immune signaling pathway.

**FIP200 activates innate immunity via RIG-I and independently of autophagy**. To determine the place of FIP200 in the linear RIG-I signaling pathway, we transfected RIG-I, MAVS, TBK1, and IRF3 together with the ISRE reporter into FIP200 wild type and knockout HEK293 cells. As shown in Fig. 5a, the ISRE activity induced by RIG-I, but not other genes, was impaired in FIP200 knockout cells, suggesting that FIP200 is upstream of MAVS. We further examined the effect of the FIP200-RIG-I interaction on poly(I:C)-induced innate immune response. We reconstituted the FIP200 knockout HEK293 cells with full-length FIP200, ATG deletion (delATG), or Claw deletion mutant (delClaw). Full-length of FIP200 rescued the phenotype of immune responses while both delATG and delClaw failed to restore IFNβ, IP10, and RANTES expression induced by poly(I:C) (Fig. 5b, Supplementary Fig. 4a, b). As ATG and Claw are required for RIG-I interaction, these data suggest that FIP200 activates innate immunity through RIG-I.

Next, we examined whether the autophagy function of FIP200 is required for RIG-I activation. We used the FIP200 autophagy defective mutant in which four critical sites for ATG13 binding are mutated into alanines (referred to as FIP200-4A)[26]. Wild-type FIP200, the autophagy-deficient mutant FIP200-4A, or the critical autophagy kinase ULK1 was transfected with RIG-I and NF-κB or ISRE reporter into HEK293 cells. Like wild-type FIP200, the FIP200-4A mutant synergistically induced both NF-κB and ISRE reporter activities with RIG-I (Supplementary Fig. 4c). By contrast, ULK1 failed to activate RIG-I (Supplementary Fig. 4c). Next, we examined the effects of the FIP200-4A mutant and ULK1 on the mRNA expression of IFNβ, IP10, and RANTES. Consistently, the FIP200-4A mutant of FIP200 synergistically induced the mRNA expression of IFNβ, IP10, and RANTES with RIG-I, whereas ULK1 was unable to activate RIG-I (Fig. 5c). We further reconstituted the FIP200 knockout HEK293 cells with the FIP200-4A mutant. The FIP200 knockout cells reconstituted with the FIP200-4A mutant fully restored IFNβ expression induced by poly(I:C) (Fig. 5d). Because ULK1 fails to activate RIG-I and the autophagy-deficient mutant of FIP200-4A is still capable of promoting RIG-I activation, it suggests that FIP200-mediated RIG-I activation is independent of the autophagy function.

**FIP200 limits RNA virus infection**. To investigate the antiviral activity of FIP200, FIP200 was transfected with RIG-I into HEK293 cells. After 24 h, cells were infected with the VSV carrying a firefly luciferase gene (VSV-Luc) for 16 h. As shown in Fig. 6a, FIP200, but not the autophagy kinase ULK1, promoted the antiviral activity of RIG-I. Next, we examined the effects of FIP200 deficiency on VSV infection in MEFs, HEK293 cells, and RAW264.7 macrophages. Viral infection activity increased approximately 4- to 12-fold in FIP200 knockout cells compared with the wild-type cells (Fig. 6b and Supplementary Fig. 5a). Furthermore, FIP200 deficiency also increased the number of infected cells (Supplementary Fig. 5b), the expression of viral proteins (Supplementary Fig. 5c), and the production of viral particles (Fig. 6c). To exclude these observations might be due to defective autophagy, we knocked out ATG5, an essential gene for the autophagy process. Contrary to the observation in FIP200 knockout cells, ATG5 deficiency impaired VSV infection

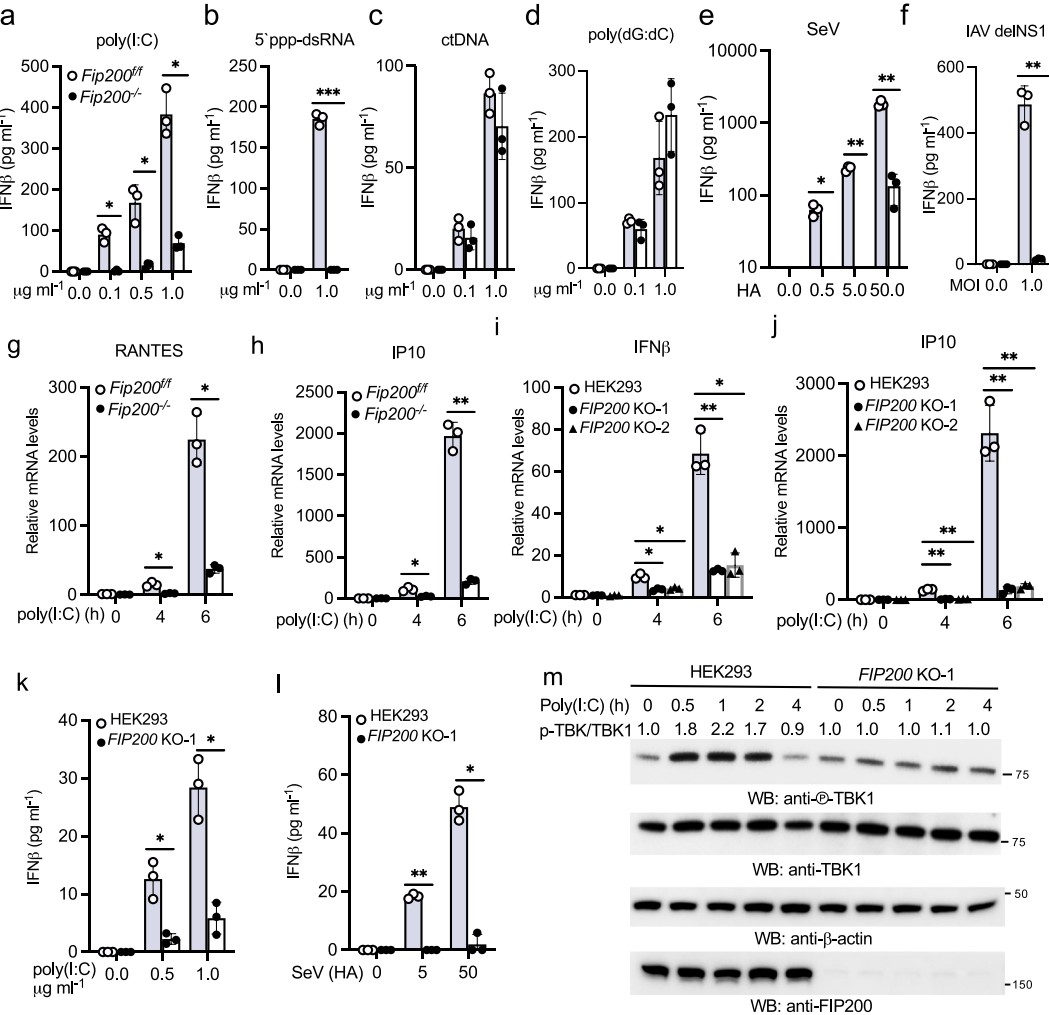

**Fig. 3 FIP200 deficiency impairs RIG-I activation in fibroblasts. a–d** Fip200[f/f] and Fip200[−/−] MEFs were transfected with indicated amount of poly(I:C) (**a**), 5′ ppp-dsRNA (**b**), calf thymus DNA (ctDNA) (**c**), or poly(dG:dC) (**d**). After 16 h, the supernatants were collected for IFNβ ELISA assays. All experiments were biologically repeated three times. Data represent means ± s.d. of three independent experiments. The P-value was calculated (two-tailed Student's t test) by comparison with the Fip200[f/f] cells. *P < 0.05, ***P < 0.001. **e**, **f** Fip200[f/f] and Fip200[−/−] MEFs were infected with Sendai virus (SeV) (**e**) or influenza A virus PR8 with NS1 deletion (IAV delNS1) (**f**). After 16 h, the supernatants were collected for IFNβ ELISA assays. All experiments were biologically repeated three times. Data represent means ± s.d. of three independent experiments. The P-value was calculated (two-tailed Student's t test) by comparison with the Fip200[f/f] cells. *P < 0.05, **P < 0.01. **g**, **h** Fip200[f/f] and Fip200[−/−] MEFs were stimulated with 1 μg ml⁻¹ poly(I:C) for indicated times. Real-time PCR was performed to determine the relative mRNA levels of RANTES (**g**) and IP10 (**h**). All experiments were biologically repeated three times. Data represent means ± s.d. of three independent experiments. The P-value was calculated (two-tailed Student's t test) by comparison with the Fip200[f/f] cells. *P < 0.05, **P < 0.01. **i**, **j** Wild type and two FIP200 knockout HEK293 cell lines were stimulated with 1 μg ml⁻¹ poly(I:C) for indicated times. Real-time PCR was performed to determine the relative mRNA levels of IFNβ (**i**) and IP10 (**j**). All experiments were biologically repeated three times . Data represent means ± s.d. of three independent experiments. The P-value was calculated (two-tailed Student's t test) by comparison with wild-type cells. *P < 0.05, **P < 0.01. **k–l** Wild type and the FIP200 knockout HEK293 cells were treated with the designated amount of poly(I:C) (**k**) or Sendai virus (**l**) for 16 h. IFNβ production was measured by ELISA. All experiments were biologically repeated three times. Data represent means ± s.d. of three independent experiments. The P-value was calculated (two-tailed Student's t test) by comparison with wild-type cells. *P < 0.05, **P < 0.01. **m** FIP200 wild type and knockout HEK293 cells were stimulated with μg ml⁻¹ poly(I:C) for indicated times. Cell lysates were collected and blotted as indicated. Band densitometry was calculated by Image J. The ratio of phosphorylated TBK1 to total TBK1 in each lane was indicated.

(Supplementary Fig. 5d), which is consistent with previous reports[27,28]. Next, we examined FIP200 knockout cells reconstituted with the FIP200-4A mutant. Although the FIP200-4A mutant is autophagy defective, it restored the antiviral activity in FIP200 knockout cells (Supplementary Fig. 5e). We further examined the effect of FIP200 deficiency on another RNA virus, HCoV OC43, which is not inhibited by autophagy[29]. In line with the results of VSV, HCoV OC43 infection activity increased approximately 10-fold in FIP200 knockout cells compared with the wild-type cells, determined by viral RNA levels (Fig. 6d) and viral titers (Fig. 6e). Thus, FIP200-mediated RIG-I activation is

independent of its autophagy function. Furthermore, we examined the role of FIP200 in DNA virus infection. The vaccinia virus (VACV) exhibited comparable infection activity in FIP200 wild type vs. knockout cells (Fig. 6f and Supplementary Fig. 5f). These data suggest that FIP200 deficiency increases host susceptibility to RNA virus, but not DNA virus.

**FIP200 is required for RIG-I activation and host defense in vivo.** The general knockout of FIP200 in mouse is embryonic lethal[30], so we adopted the Fip200[f/f];LysM-Cre mice for conditional knockout in myeloid cells. Western blot analysis found

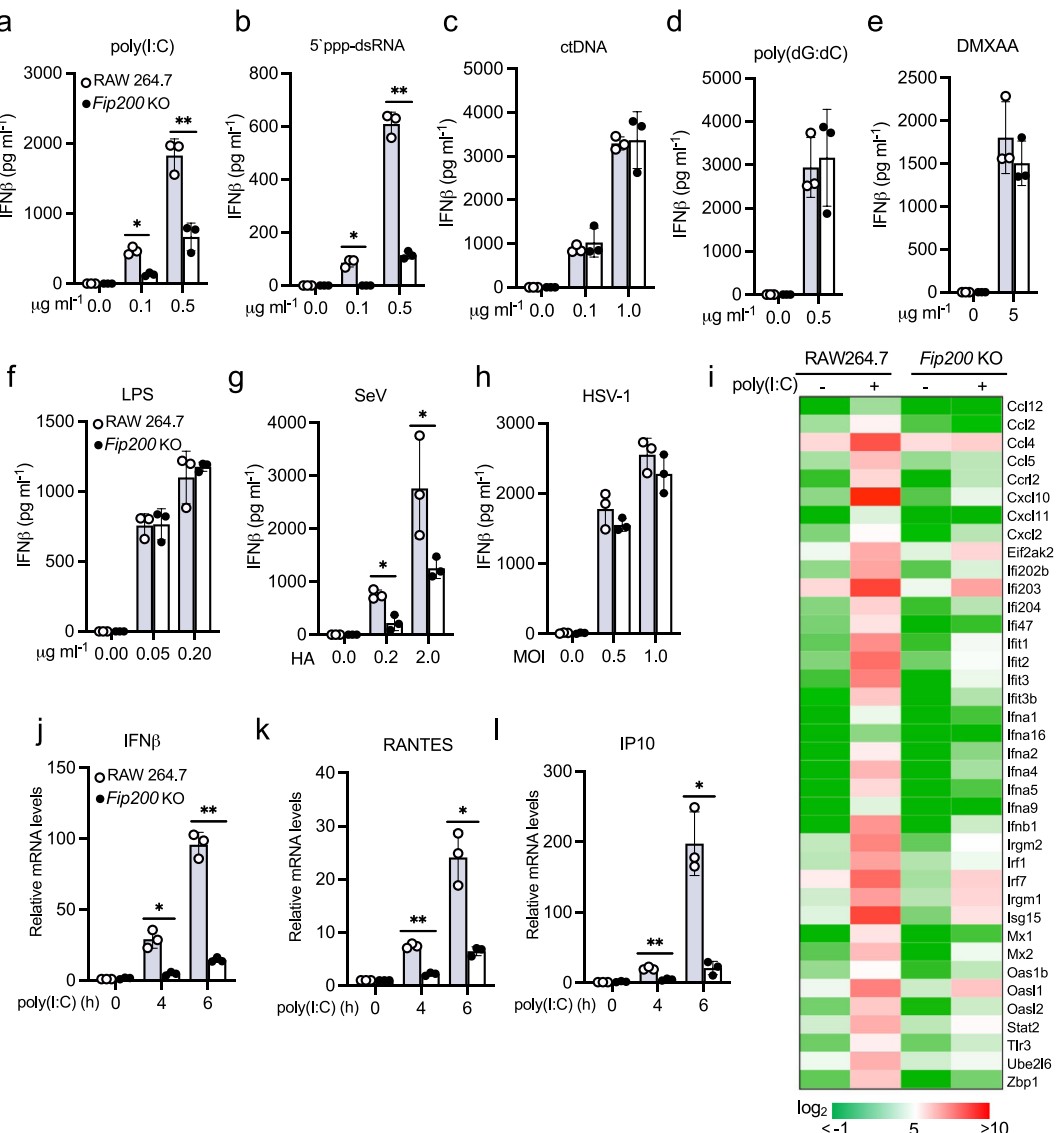

**Fig. 4 FIP200 deficiency impairs RIG-I activation in macrophages. a–f** Wild type and FIP200 knockout RAW264.7 macrophages were stimulated with the indicated amount of poly(I:C) (**a**), 5' ppp-dsRNA (**b**), calf thymus DNA (ctDNA) (**c**), poly(dG:dC) (**d**), DMXAA (**e**), or LPS (**f**). After 16 h, the supernatants were collected for IFNβ ELISA assays. All experiments were biologically repeated three times. Data represent means ± s.d. of three independent experiments. The P-value was calculated (two-tailed Student's t test) by comparison with wild-type cells. *P < 0.05, **P < 0.01. **g, h** Wild type and the FIP200 knockout RAW264.7 macrophages were infected with SeV (**g**) or HSV-1 d109 mutant virus (**h**). After 16 h, the supernatants were collected for IFNβ ELISA assays. All experiments were biologically repeated three times. Data represent means ± s.d. of three independent experiments. The P-value was calculated (two-tailed Student's t test) by comparison with wild-type cells. *P < 0.05. **i** Heatmap of the RNA sequencing results of wild type and FIP200 knockout RAW264.7 macrophages stimulated with 1 µg ml$^{-1}$ poly(I:C) for 4 h. **j–l** Wild type and FIP200 knockout RAW264.7 macrophages were stimulated with 1 µg ml$^{-1}$ poly(I:C) for indicated times. Real-time PCR was performed to determine the relative mRNA levels of IFNβ (**j**), RANTES (**k**), and IP10 (**l**). All experiments were biologically repeated three times. Data represent means ± s.d. of three independent experiments. The P-value was calculated (two-tailed Student's t test) by comparison with wild-type cells. *P < 0.05, **P < 0.01.

more than 90% knockdown of FIP200 expression in the bone marrow-derived macrophages (BMDMs) of *Fip200$^{f/f}$*;LysM-Cre mice (Fig. 7a). Innate immune responses to poly(I:C) were impaired in the BMDMs of *Fip200$^{f/f}$*;LysM-Cre mice (Fig. 7a, Supplementary Fig. 6a, b). Furthermore, we examined the in vivo innate immune response by injection of PEI-poly(I:C). Consistently, IFNβ production in the blood dramatically reduced in *Fip200$^{f/f}$*;LysM-Cre mice (Fig. 7b). These combined data suggest the deficiency of FIP200 impairs host innate immune response to cytosolic dsRNA in vivo.

Next, we infected *Fip200$^{f/f}$* and *Fip200$^{f/f}$*;LysM-Cre mice with VSV intravenously. As shown in Fig. 7c, the *Fip200$^{f/f}$*;LysM-Cre

mice displayed increased susceptibility to VSV. About 50% of *Fip200$^{f/f}$*;LysM-Cre mice became moribund compared with approximately 10% in wild-type group. ELISA assays further found that a high amount of IFNβ was produced in the blood of *Fip200$^{f/f}$* mice one day post of infection (Fig. 7d). By contrast, only little IFNβ was detected in the blood of *Fip200$^{f/f}$*;LysM-Cre mice (Fig. 7d). Consistently, *Fip200$^{f/f}$*;LysM-Cre mice exhibited higher viral load compared with *Fip200$^{f/f}$* mice (Fig. 7e). Following VSV infection, *Fip200$^{f/f}$*;LysM-Cre mouse exhibited a marked diffuse interstitial pneumonia with focal areas of vasculitis and bronchitis with mixed polynuclear/mononuclear infiltrates and scattered areas of epithelial necrosis whereas lung

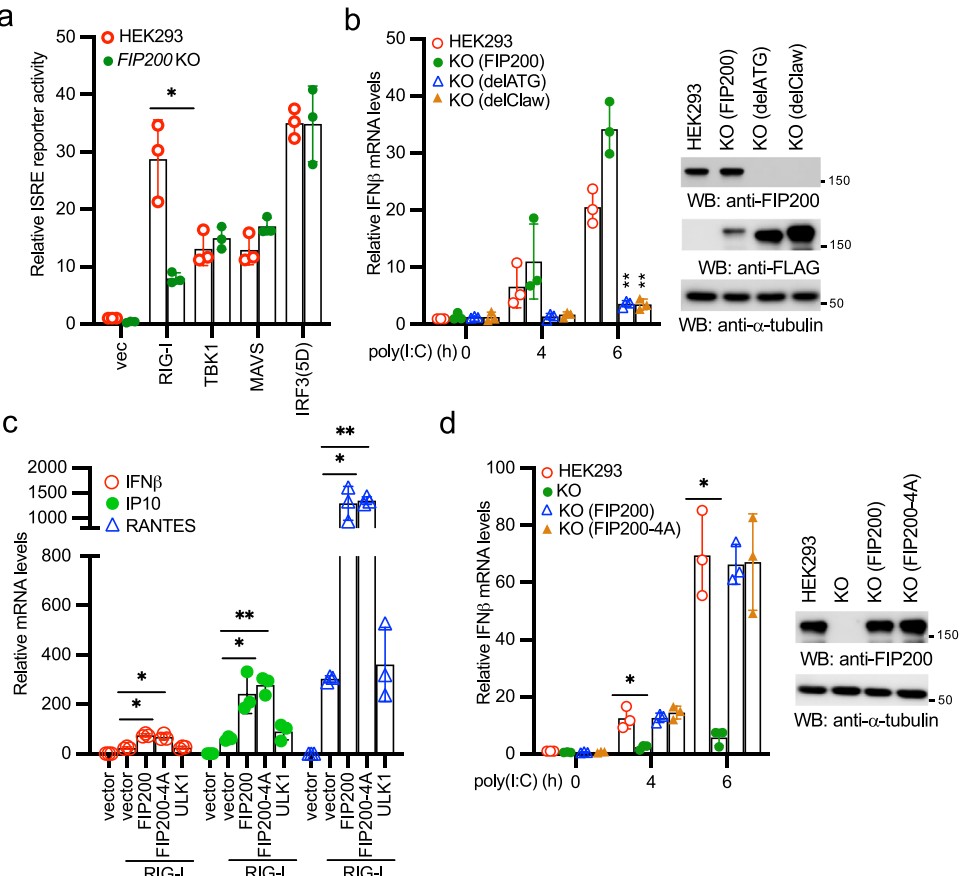

**Fig. 5 FIP200 activates innate immunity via RIG-I but independently of its autophagy function. a** FIP200 wild type and knockout HEK293 cells were transfected with vector, RIG-I, TBK1, MAVS, or IRF3(5D) together with pRL-SV40 and pISRE-Luc. After 48 h, cells were collected and the ratio of firefly luciferase to *Renilla* luciferase was calculated to determine the relative activity of IFN reporter. All experiments were biologically repeated three times. Data represent means ± s.d. of three independent experiments. The *P*-value was calculated (two-tailed Student's *t* test) by comparison with HEK293 cells. \**P* < 0.05. **b** Wild-type HEK293 cells, FIP200 knockout cells reconstituted with full-length FIP200, delATG, or delClaw were stimulated with 1 μg ml⁻¹ poly(I:C) for designated times. Then, RNA was extracted, and real-time PCR for IFNβ was performed. All experiments were biologically repeated three times. Data represent means ± s.d. of three independent experiments. (\*\**P* < 0.01 vs. the wild-type cells by two-tailed Student's *t* test). Right panel shows the expression of FIP200 and the mutants in the reconstituted cells. The anti-FIP200 antibody detects the C-terminal end of FIP200 and cannot recognize delATG and delClaw. **c** RIG-I was transfected with vector, FIP200, FIP200-4A, or ULK1 into HEK293 cells. After 48 h, RNA was extracted, and real-time PCR assays for IFNβ, IP10, and RANTES were performed. All experiments were biologically repeated three times. Data represent means ± s.d. of three independent experiments. (\**P* < 0.05, \*\**P* < 0.01, by two-tailed Student's *t* test). **d** Wild-type HEK293 cells, FIP200 knockout cells, and FIP200 knockout cells reconstituted with full-length FIP200 or the 4A mutant were stimulated with 1 μg ml⁻¹ poly(I:C) for designated times. Then, RNA was extracted, and real-time PCR for IFNβ was performed. All experiments were biologically repeated three times. Data represent means ± s.d. of three independent experiments. (\**P* < 0.05 vs. the wild-type cells by two-tailed Student's *t* test). Right panel shows the expression of FIP200 and *th*e mutant in the reconstituted cells.

tissue obtained from *Fip200^{f/f}* mouse was essentially normal. (Supplementary Fig. 6c).

Coronaviruses are single-stranded positive-sense RNA viruses and sensed by the RIG-I-like receptor, including RIG-I and MDA5[31,32]. Thus, we also infected *Fip200^{f/f}* and *Fip200^{f/f}*;LysM-Cre suckling mice with HCoV OC43 intracerebrally. About 80% of *Fip200^{f/f}*;LysM-Cre mice became moribund compared with approximately 30% in wild-type group (Fig. 7f). Similar to VSV, HCoV OC43 induced little IFNβ (Fig. 7g) but higher viral load (Fig. 7h) in *Fip200^{f/f}*;LysM-Cre mice than wild-type mice. Overall, these data suggest FIP200 is required for host innate immune defense to RNA viruses in vivo.

**FIP200 facilitates the de-repression and dimerization of 2CARD.** RIG-I activation involves several sequential steps, including the release of 2CARD, ubiquitination, and oligomerization. Since FIP200 interacts with the 2CARD of RIG-I, we

hypothesized that FIP200 competed with the Hel-CTD of RIG-I for 2CARD binding. As shown in Fig. 8a, the competition immunoprecipitation found that FIP200 blocked the interaction between the 2CARD and the Hel-CTD of RIG-I. By contrast, the RIG-I binding deficient mutant, delClaw, failed to interfere with the interaction (Fig. 8a). In vitro binding competition assays further demonstrated that the ATG or the Claw domain of FIP200 alone was sufficient to block the interaction between the 2CARD and the Hel-CTD of RIG-I (Fig. 8b, Supplementary Fig. 7a). A previous study found that several sites (R109, E113, R117, and L185) in the second CARD of RIG-I are responsible for the interaction with the helicase domain[33]. Next, we examined whether these sites were also required for the interaction with FIP200. As shown in Fig. 8c, sites E113, R117, and L185, but not R109, were required for the interaction between FIP200 and RIG-I, suggesting that FIP200 competes with the helicase domain on these sites.

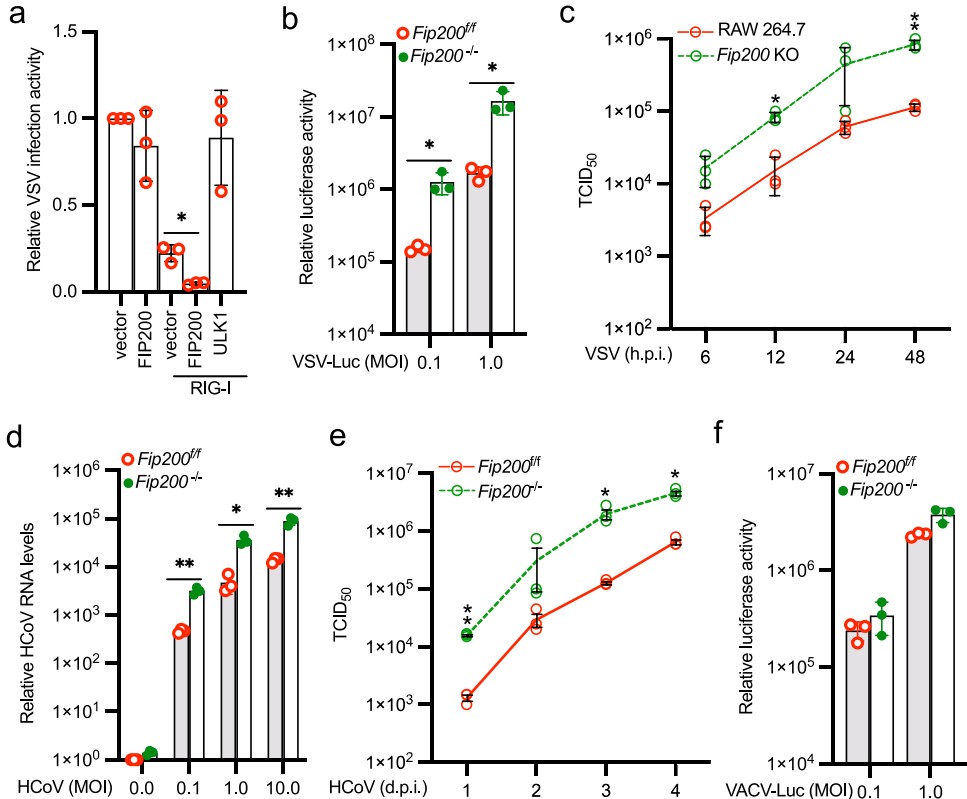

**Fig. 6 FIP200 deficiency increases host susceptibility to RNA virus infection. a** RIG-I was transfected with vector, FIP200, or ULK1 into HEK293 cells. After 24 h, cells were infected with the VSV carrying a luciferase gene (VSV-Luc) for 16 h. Luciferase activity was measured to determine relative viral infection activity. All experiments were biologically repeated three times. Data represent means ± s.d. of three independent experiments. (*$P < 0.05$, by two-tailed Student's $t$ test). **b** $Fip200^{f/f}$ and $Fip200^{-/-}$ MEFs were infected with VSV-Luc for 16 h. Luciferase activity was measured to determine relative viral infection activity. All experiments were biologically repeated three times. Data represent means ± s.d. of three independent experiments. The $P$-value was calculated (two-tailed Student's $t$ test) by comparison with wild-type cells. *$P < 0.05$. **c** Wild type and FIP200 knockout RAW 264.7 macrophages were infected with 0.001 MOI of VSV. After the designated hour post-infection (h.p.i.), virus titers were determined by TCID$_{50}$ in Vero cells. All experiments were biologically repeated three times. Data represent means ± s.d. of three independent experiments. The $P$-value was calculated (two-tailed Student's $t$ test) by comparison with the wild-type cells. *$P < 0.05$, **$P < 0.01$. **d** $Fip200^{f/f}$ and $Fip200^{-/-}$ MEFs were infected with the indicated MOI of HCoV OC43. After 24 h, the RNA levels of viral membrane gene were determined by qPCR. All experiments were biologically repeated three times. Data represent means ± s.d. of three independent experiments. The $P$-value was calculated (two-tailed Student's $t$ test) by comparison with the wild-type cells. *$P < 0.05$, **$P < 0.01$. **e** $Fip200^{f/f}$ and $Fip200^{-/-}$ MEFs were infected with 0.1 MOI of HCoV OC43. After the designated day post-infection (d.p.i.), virus titers were determined by TCID$_{50}$ in Vero cells. All experiments were biologically repeated three times. Data represent means ± s.d. of three independent experiments. The $P$-value was calculated (two-tailed Student's $t$ test) by comparison with the wild-type cells. *$P < 0.05$, **$P < 0.01$. **f** $Fip200^{f/f}$ and $Fip200^{-/-}$ MEFs were infected with the VACV carrying a luciferase gene (VACV-Luc) for 16 h. Luciferase activity was measured to determine relative viral infection activity. All experiments were biologically repeated three times. Data represent means ± s.d. of three independent experiments.

Next, we examined the role of FIP200 in RIG-I ubiquitination. First, we examined the effect of ectopic expression of FIP200 on RIG-I ubiquitination. FIP200 or control vector was co-transfected with RIG-I and ubiquitin into HEK293 cells. Immunoprecipitation showed that FIP200 had little impact on RIG-I ubiquitination (Supplementary Fig. 7b). We further transfected RIG-I and ubiquitin into wild type and FIP200 knockout HEK293 cells. Similarly, deficiency of FIP200 has a marginal effect on RIG-I ubiquitination (Supplementary Fig. 7c). These data suggest that FIP200 might not be involved in the RIG-I ubiquitination process.

As reported previously[34], the 2CARD failed to pull down itself by co-IP (Fig. 8d). However, FIP200 enhanced 2CARD dimerization (Fig. 8d). Recent structural study found FIP200 forms a dimer through the ATG domain[23]. Therefore, it is plausible that FIP200 facilitates RIG-I dimerization or oligomerization. We first examined RIG-I oligomerization in FIP200 wild type and knockout cells. Sucrose gradient fractionation assays found that poly(I:C) induced RIG-I to distribute in high molecular weight fractions in wild-type cells but not in FIP200 knockout cells (Supplementary Fig. 7d), suggesting that FIP200 is required for RIG-I oligomerization. We hypothesized that FIP200 formed a dimer, thereby facilitating RIG-I dimerization and oligomerization. Thus, we examined whether FIP200 formed a dimer. Consistent with the previous report, the ATG domain alone was sufficient to form a dimer (Supplementary Fig. 7e). CC domain is known to form a dimer and oligomer[35]. Co-IP found the CC of FIP200 also formed a dimer (Supplementary Fig. 7f). Next, we examined whether these domains affected FIP200 dimerization. Deletion of either the CC domain (delCC) or ATG (delATG) reduced the dimerization of FIP200 (Supplementary Fig. 7g). Consistently, delCC or delATG impaired FIP200-mediated 2CARD dimerization (Fig. 8e). Since ATG is required for RIG-I interaction, we used the delCC mutant to examine the effect of FIP200 dimerization deficiency on RIG-I activation. The delCC was reconstituted in FIP200 knockout HEK293 cells.

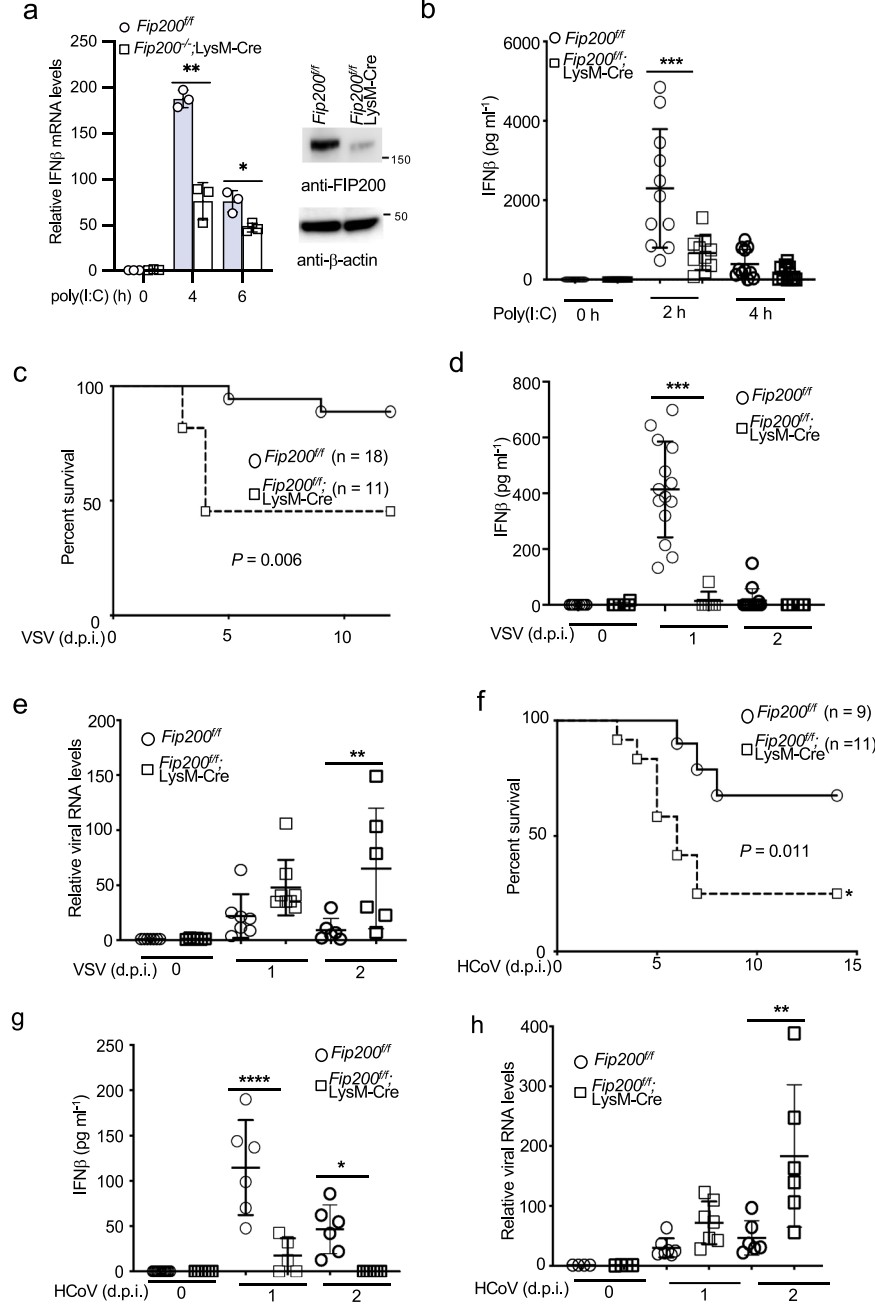

**Fig. 7 FIP200 is required for innate host defense to RNA virus infection in vivo. a** The BMDMs of *Fip200*^f/f and *Fip200*^f/f;LysM-Cre mice were stimulated with 1 μg ml$^{-1}$ of poly(I:C) for indicated times. Real-time PCR was performed to determine the relative IFNβ mRNA levels. Data represent means ± s.d. of three independent experiments. The *P*-value was calculated (two-tailed Student's *t* test) by comparison with the wild-type cells. *$P < 0.05$, **$P < 0.01$. The right panel shows the knockout efficiency by Western blotting. **b** *Fip200*^f/f and *Fip200*^f/f;LysM-Cre mice were injected with 50 μg of poly(I:C) plus 6 μl of in vivo-jetPEI via tail vein for designated times. Blood was collected for IFNβ ELISA assay. ***$P < 0.001$, by one-way ANOVA, followed by Tukey's multiple comparison test. **c** Survival curve of VSV-infected *Fip200*^f/f and *Fip200*^f/f;LysM-Cre mice. Eighteen *Fip200*^f/f mice and eleven *Fip200*^f/f;LysM-Cre mice were infected with 2 × 10$^5$ TCID$_{50}$ of VSV intravenously. Mice were monitored at the indicated d.p.i. $P = 0.006$, calculated by Mantel–Cox log-rank test.
**d** *Fip200*^f/f and *Fip200*^f/f;LysM-Cre mice were infected with 2 × 10$^5$ TCID$_{50}$ of VSV intravenously. Blood was collected for IFNβ ELISA assays at 0, 1, and 2 d.p.i. ***$P < 0.001$, by one-way ANOVA, followed by Tukey's multiple comparison test. **e** *Fip200*^f/f and *Fip200*^f/f;LysM-Cre mice were infected with 2 × 10$^5$ TCID$_{50}$ of VSV intravenously. Brain tissues were collected for RNA extract. VSV RNA levels were determined by real-time PCR. **$P < 0.01$, by one-way ANOVA, followed by Sidak's multiple comparisons test. **f** Survival curve of HCoV OC43-infected *Fip200*^f/f and *Fip200*^f/f;LysM-Cre mice. Nine *Fip200*^f/f mice and elven *Fip200*^f/f;LysM-Cre mice were infected with 1 × 10$^5$ TCID$_{50}$ of HCoV OC43 intracerebrally. Mice were monitored at the indicated day post-infection (d.p.i.). $P = 0.011$, calculated by Mantel–Cox log-rank test. **g** *Fip200*^f/f and *Fip200*^f/f;LysM-Cre mice were infected with 1 × 10$^5$ TCID$_{50}$ of HCoV OC43 intravenously. Blood was collected for IFNβ ELISA assays at 0, 1, and 2 d.p.i. *$P < 0.05$, ****$P < 0.0001$, by one-way ANOVA, followed by Tukey's multiple comparison test. **h** *Fip200*^f/f and *Fip200*^f/f;LysM-Cre mice were infected with 1 × 10$^5$ TCID$_{50}$ of HCoV OC43 intravenously. Brain tissues were collected for RNA extract. HCoV RNA levels were determined by real-time PCR. **$P < 0.01$, by one-way ANOVA, followed by Tukey's multiple comparison test.

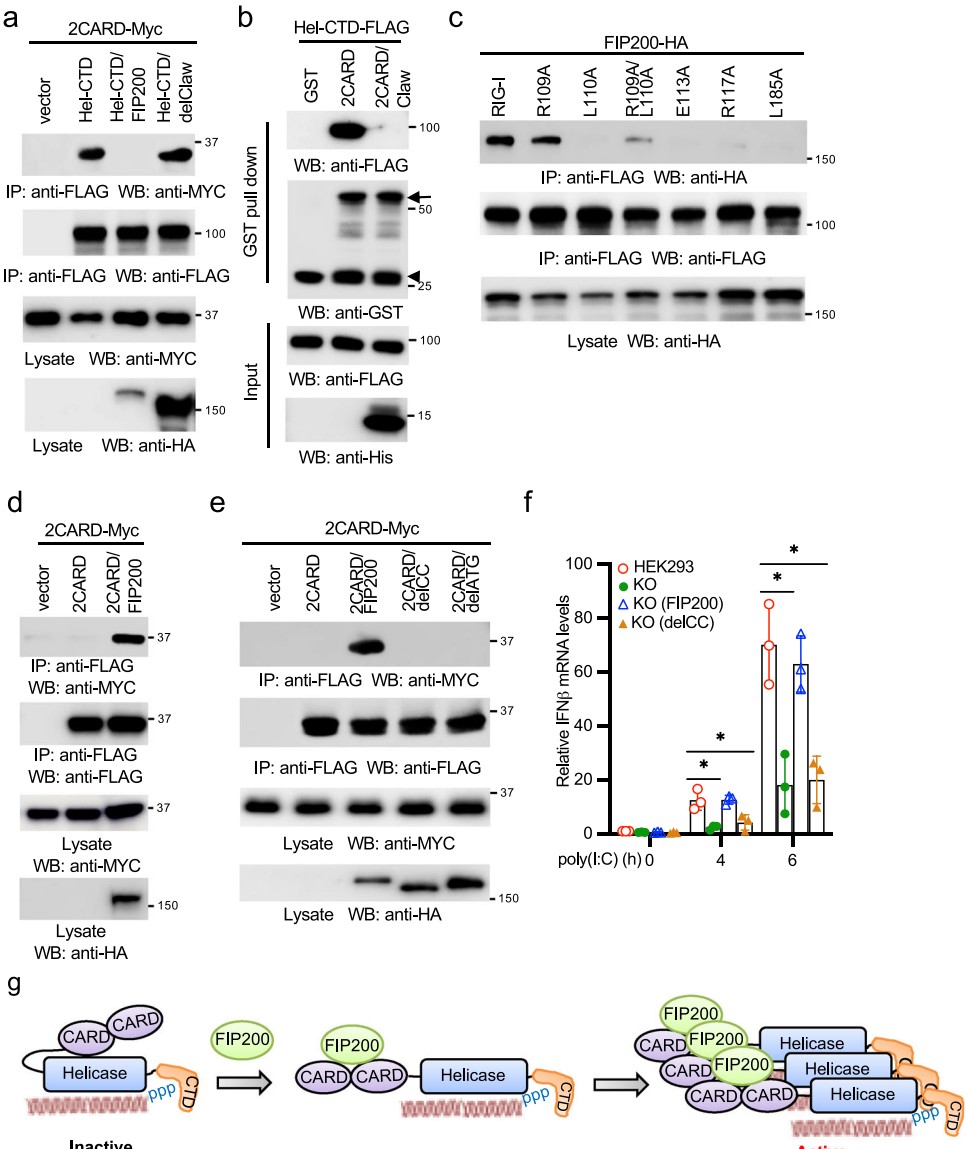

**Fig. 8 FIP200 facilitates the release and oligomerization of 2CARD. a** Myc-tagged 2CARD (2CARD-Myc) was co-transfected with vector, the FLAG-tagged Hel-CTD, Hel-CTD plus HA-tagged FIP200, or Hel-CTD plus HA-tagged delClaw into HEK293 cells. After 48 h, cell lysates were immunoprecipitated and blotted as indicated. **b** Purified recombinant FLAG-tagged Hel-CTD (Hel-CTD-FLAG) was mixed with GST, GST-tagged 2CARD (2CARD-GST) or 2CARD-GST plus His-tagged Claw at 4 °C. After 16 h, GST pull-down assay was performed. Arrow indicates the 2CARD-GST. Arrowhead indicates the cleaved GST from 2CARD-GST during extraction and purification from *E. coli*. **c** FIP200-HA was co-transfected with wild-type RIG-I or the indicated point mutant into HEK293 cells. After 48 h, cell lysates were immunoprecipitated and blotted as indicated. **d** 2CARD-Myc was co-transfected with vector, FLAG-tagged 2CARD (2CARD-FLAG), 2CARD-FLAG plus FIP200-HA into HEK293 cells. After 48 h, cell lysates were immunoprecipitated and blotted as indicated. **e** 2CARD-Myc was co-transfected with vector, 2CARD-FLAG, 2CARD-FLAG plus FIP200-HA, delCC-HA or delATG-HA into HEK293 cells. After 48 h, cell lysates were immunoprecipitated and blotted as indicated. **f** Wild-type HEK293 cells, FIP200 knockout cells, FIP200 knockout cells reconstituted with full-length FIP200 or the delCC mutant were stimulated with 1 μg ml⁻¹ poly(I:C) for designated times. RNA was extracted and real-time PCR for IFNβ was performed. All experiments were biologically repeated three times. Data represent means ± s.d. of three independent experiments. The *P*-value was calculated (two-tailed Student's *t* test) by comparison with HEK293 cells. *$P < 0.05$. **g** Model of FIP200-mediated RIG-I activation.

However, the delCC failed to restore IFNβ, IP10, and RANTES expression induced by poly(I:C) (Fig. 8f; Supplementary Fig. 7h, i), suggesting that FIP200 dimerization is required for RIG-I activation. The combined data suggest that FIP200 potentiates RIG-I activation by facilitating the release of 2CARD and the dimerization of 2CARD (Fig. 8g).

## Discussion

Here, we provide multiple lines of evidence to define FIP200 as a positive regulator of RIG-I. First, dsRNA induces FIP200-RIG-I interaction. FIP200 interacts with the CARD of RIG-I through its C-terminal Claw domain. Second, dsRNA-induced RIG-I activation is impaired in multiple FIP200 deficiency cell lines, including MEF, HEK293, RAW264.7, and BMDM. Third, ablation of FIP200 increases host susceptibility to RNA viruses, VSV and HCoV, but not the DNA virus VACV. Fourth, FIP200 is required for RIG-I activation and host defense to VSV and HCoV in vivo. Last, FIP200 facilitates RIG-I 2CARD release and dimerization, thereby promoting RIG-I activation.

Our data demonstrate that FIP200-mediated RIG-I activation is independent of autophagy based on the following evidence.

First, to exclude potential involvement of autophagy, we used the FIP200 autophagy defective mutant, FIP200-4A, which fails to bind ATG13. Our data demonstrate that the FIP200-4A mutant is able to activate RIG-I. Second, it has been shown that ATG5 deficiency leads to the induction of type I IFN expression in MEFs and the inhibition of VSV infection[27,28]. By contrast, our data showed that FIP200 knockout abrogated type I IFN expression and increased VSV infection, suggesting RIG-I mediated IFN signaling has much stronger impact on host defense than autophagy in MEFs. Third, it has been shown autophagy either does not inhibit or has little effect on coronaviruses[29,36]. Our data found that FIP200 deficiency increased HCoV OC43 infection in cells and mice. Taken together, our data suggest that FIP200 is critical for RIG-I activation and suppression of RNA virus infection.

Recent studies revealed the structure-based activation of RIG-I[37–39]. The structure of inactive duck full-length RIG-I shows that in the pre-dsRNA binding state, the 2CARD is sequestered through the interaction of CARD2 with the insertion domain Hel2i of the helicase domain[37]. The structure of the RNA-bound Hel-CTD showed a closed conformation, suggesting that the conformation switch of helicase expels 2CARD. A recent study using FRET demonstrated that dsRNA binding induces the dissociation between the 2CARD and helicase domain[40]. It is believed the expel frees and exposes the 2CARD for interactions and signaling. Biochemistry evidence largely agrees with the structural studies. Biochemical studies also showed that ubiquitination of CTD is required for the release of autorepression of RIG-I[41]. However, it is not clear whether other host factors regulate the release of 2CARD. Our study shows that FIP200 binds the 2CARD of RIG-I and facilitates the CARD's dissociation with the helicase domain upon RNA binding. Interestingly, FIP200 competes with several, but not all of the binding sites between the CARD2 and the helicase. It suggests that FIP200 might not be able to release the 2CARD from the helicase domain without dsRNA binding due to lower affinity. After dsRNA binding, the switch of the conformation of helicase domain exposes the 2CARD. Our data showed that poly(I:C) induced the interaction between RIG-I and FIP200, suggesting that FIP200 interacts with the exposed 2CARD. We speculate the binding of FIP200 might facilitate the process of CARD expel and sustain the released status of the 2CARD, thereby promoting the activity of RIG-I.

The release of 2CARD is required; however, it is not sufficient for RIG-I activation. To be activated, RIG-I needs to form an oligomer or a filament. RIG-I multimerizes along the dsRNA; however, the bead-like multimer cannot activate RIG-I. Several studies also showed that the short (10–24 bp) phosphorylated synthetic RNA duplexes potently activated RIG-I signaling in vitro and in vivo[42,43]. These short RNA ligands can only bind a single RIG-I molecule; therefore, RIG-I oligomerization can be RNA independent, suggesting RIG-I oligomerizes through its domain or post-translational modifications. Recent studies show that the K63-linked polyubiquitination mediates oligomerization of RIG-I. Three ubiquitin E3 ligases, including TRIM25[8], MEX3C[9], and TRIM4[11], have been reported to activate RIG-I signaling by K63-linked polyubiquitination of the 2CARD. TRIM25 was first found to bind the 2CARD and mediate polyubiquitination of 2CARD at Lys172[8]. The other two studies showed that MEX3C targets RIG-I for ubiquitination at Lys48, 99, and 169[9] while TRIM4 ubiquitinates RIG-I at Lys154, 164, and 172[11]. However, recent studies using knockout cell lines and knockout mice showed that another ubiquitin E3 ligase, RIPLET, is the predominant E3 ligase for RIG-I K63-linked ubiquitination and activation in both human and mouse[44–46]. Unlike other E3 ligases, RIPLET ubiquitinates the CTD domain of RIG-I at Lys849 and Lys851[10]. Interestingly, unanchored K63-linked

polyubiquitin is also found to non-covalently bind the 2CARD and promote CARD tetramerization and concomitant signal activation[14]. A further structural study showed that the 2CARD are stabilized into a tetrameric 'lock-washer' conformation in which three chains of K63-linked di-ubiquitin are bound along the outer rim of the helical trajectory to assemble a stable CARD tetramer[13]. It is plausible that RIPLET produces free polyubiquitin in cells upon viral infection or ligand stimulation, but it needs further investigation

Besides the K63-linked polyubiquitination, other regulatory mechanisms for tetramerization/ oligomerization of the 2CARD are limited. For example, RIPLET dimerization has been shown to promote RIG-I oligomerization and filamentation[44]; however, RIPLET binds the helicase domain[47], suggesting that RIPLET dimerization might not directly induce CARD oligomerization. Our study showed that FIP200 interacted with the 2CARD and enhanced CARD dimerization. We and others found that FIP200 formed dimers through its domains, CC and ATG. The ATG domain forms a Claw shape dimer[23]. The CC domains are known to form when two or more α-helices self-assemble by winding around each other to form a left-handed supercoil. Dimers, trimers, and tetramers are the most common CC structures[35]. The CC or ATG alone is sufficient to form a dimer. However, our data found that both CC and ATG are required for FIP200 dimerization. As illustrated in Fig. 8g, we propose a model that FIP200 dimerization facilitates the dimerization/oligomerization of 2CARD. Although our data show that FIP200 has little effect on RIG-I ubiquitination, we cannot discriminate the cognate ubiquitination between the 2CARD and the CTD domain due to multiple ubiquitination sites on each domain. It is also not clear how FIP200 co-opt with the conjugated or bound-free polyubiquitin on RIG-I activation. We speculate both regulatory mechanisms are required for the full activity and sustained activation of RIG-I. Future studies will explore the crosstalk between both mechanisms.

In summary, we have demonstrated that FIP200 is indispensable for RIG-I activation and innate antiviral response in vitro and in vivo. These findings not only elucidate a regulatory mechanism for RIG-I activation but also provide insights on the design of potential therapeutics for autoimmune and infectious diseases.

## Methods

**Mice.** All activities described here were performed in accordance with the guidelines put forth in the Guide for the Care and Use of Laboratory Animals and were approved by the Institutional Animal Care and Use Committee of Oklahoma State University (protocol VM-17-33) and Tulane University (protocol 826). The *Fip200^{f/f}*;LysM-Cre mouse strain was a generous gift from Dr. Herbert W. Virgin (Washington University, St. Louis, MO). Cohorts of age-matched (6–8-week old) mice were injected with the virus in 100 μl sterile PBS via the tail vein. Body weights were recorded, and signs of illness were monitored daily. Mice with 25% body weight loss were considered moribund and euthanized. For histopathology, tissues from infected mice were fixed in formalin for 24 h before being transferred to 70% ethanol. Specimens were paraffin-embedded and stained by the Histopathology Core services.

Poly(I:C) were complexed to in vivo-jetPEI (Polyplus-transfection) with a N/P (nitrogen/phosphate) ratio of 6 according to the manufacturer's instructions. Unless otherwise indicated, 50 μg of poly(I:C) plus 6 μl of in vivo-jetPEI was injected per mouse via the tail vein. At various time points, sera were collected for IFNβ ELISA.

**Cells**. HEK293 cells (ATCC, # CRL-1573), RAW 264.7 (ATCC, # TIB-71), Vero cells (ATCC, # CCL-81), L929 cells (ATCC, # CCL-1), and MEFs were maintained in Dulbecco's Modified Eagle Medium (Life Technologies, # 11995-065) containing antibiotics (Life Technologies, # 15140-122) and 10% fetal bovine serum (Life Technologies, # 26140-079). A549 cells (ATCC, # CCL-185) were cultured in RPMI Medium 1640 (Life Technologies, # 11875-093) plus 10% fetal bovine serum and 1 × MEM Non-Essential Amino Acids Solution (Life Technologies, # 11140-050).

To generate BMDMs, we isolated and cultured mouse bone marrow cells for 7 days in DMEM supplemented with 30% of L929 cell supernatant as a source of GM-CSF.

**Viruses.** VSV Indiana strain was purchased from ATCC (# VR-1238). The VSV carrying a luciferase gene (VSV-Luc) and the VSV expressing a GFP gene (VSV-GFP) were kind gifts from Dr. Sean Whelan (Harvard Medical School, MA). Viral titration was performed as the following. Vero cells were infected with a serial diluted VSV. After 1 h, the medium was removed and replaced by the DMEM plus 5% FBS and 1% agarose. After 3 d, cells were examined for cytopathic effects to determine $TCID_{50}$ or were fixed using the methanol–acetic acid (3:1) fixative and stained using a Coomassie blue solution to determine MOI.

HCoV OC43 was purchased from ATCC (# VR-1558). Sendai virus was purchased from Charles River (Wilmington, MA). HSV-1 d109 was a gift from Dr. Neal DeLuca (University of Pittsburg, Pittsburg, PA). VACV with Luc gene (VACV-Luc) was a gift from Dr. David L. Bartlett (University of Pittsburg, Pittsburg, PA). IAV delNS1 was a gift from Dr. Adolfo Garcia-Sastre (Mount Sinai School of Medicine, NY).

**Plasmids.** RIG-I, FIP200, and the FIP200-4A mutant constructs were reported previously[26,48]. pMXs-IP-EGFP-hFIP200 was a gift from Dr. Noboru Mizushima (Addgene, plasmid # 38192). HA-ULK1 was a gift from Dr. Do-Hyung Kim (Addgene, plasmid # 31963).

**Antibodies.** Primary antibodies: Anti-β-actin [Abcam, # ab8227, WB (1:1,000)], anti-FLAG [Sigma, # F3165, WB (1:1000), IFA (1:100)], anti-ubiquitin [Cell Signaling Technology, # 3933S, WB (1:1,000)], anti-HA [Cell Signaling Technology, # 3724, WB (1:1,000), IFA (1:100)], anti-RIG-I [Santa Cruz Biotechnology, # sc-376845, WB (1:1,000)], anti-RIG-I [Novus Biologicals, # NBP2-61849, IFA (1:100)], anti-GFP [Santa Cruz Biotechnology, # sc-9996, WB (1:2,000)], anti-FIP200 [Cell Signaling Technology, # 12436S, WB (1:1,000)], anti-FIP200 [Millipore, # MABC128, IFA (1:100)], anti-TBK1 [Cell Signaling Technology, # 3504S, WB (1:1,000)], anti- phospho-TBK1 (Ser172) [Cell Signaling Technology, # 5483S, WB (1:1,000)], anti-human MAVS [Cell Signaling Technology, # 24930S, WB (1:1,000)], anti-mouse MAVS [Cell Signaling Technology, #4983S, WB (1:1,000)], anti-IRF3 [Bethyl Laboratories, # A303-384A, WB (1:1,000)], anti-GST [Cell Signaling Technology, # 2624S, WB (1:1,000)], anti-His [ThermoFisher Scientific, # MA1-21315, WB (1:1,000)], anti-MBP [New England Biolabs, # E8032L, WB (1:1,000)], anti-Myc [Cell Signaling Technology, # 2276S, WB (1:1,000)], anti-Myc [Bethyl Laboratories, # A190-105A, WB (1:1,000)].

Secondary antibodies: Goat anti-Mouse IgG-HRP [Bethyl Laboratories, # A90-116P, WB (1:10,000)], Goat anti-Rabbit IgG-HRP [Bethyl Laboratories, # A120-201P, WB (1:10,000)], Alexa Fluor 594 Goat Anti-Mouse IgG (H + L) [Life Technologies, # A11005, IFA (1:200)], Alexa Fluor 488 Goat Anti-Rabbit IgG (H + L) [Life Technologies, # A11034, IFA (1:200)].

**Ligands.** Poly(I:C) LWM (Invivogen, # tlrl-picw), Poly(I:C) HWM (Invivogen, # tlrl-pic), Poly(A:U) (Invivogen, # tlrl-pau), LPS (Invivogen, # tlrl-pb5lps), 5′-ppp-dsRNA (Invivogen, # tlrl-3prna), poly(dA:dT) (Invivogen, # tlrl-patn), poly(dG:dC) (Invivogen, # tlrl-pgcn), ctDNA (Sigma, # D1501). For stimulation in cells, all ligands except LPS and poly(A:U) were transfected into cells using polyethylenimine (PEI). LPS was added directly to the cells.

**Sample preparation, Western blotting, and immunoprecipitation.** Approximately $1 \times 10^6$ cells were lysed in 500 μl of tandem affinity purification (TAP) lysis buffer [50 mM Tris-HCl (pH 7.5), 10 mM $MgCl_2$, 100 mM NaCl, 0.5% Nonidet P40, 10% glycerol, the Complete EDTA-free protease inhibitor cocktail tablets (Roche, # 11873580001)] for 30 min at 4 °C. The lysates were then centrifuged for 30 min at 15,000 × rpm. Supernatants were collected and mixed with the Lane Marker Reducing Sample Buffer (Thermo Fisher Scientific, # 39000).

Western blotting and immunoprecipitation were performed as the following. Samples (10–15 μl) were loaded into Mini-Protean TGX Precast Gels, 15 well (Bio-Rad, # 456-103), and run in 1 × Tris/Glycine/SDS Buffer (Bio-Rad, # 161-0732) for 35 min at 200 V. Protein samples were transferred to Immun-Blot PVDF Membranes (Bio-Rad, # 162-0177) in 1 × Tris/Glycine Buffer (Bio-Rad, # 161-0734) at 70 V for 60 min. PVDF membranes were blocked in 1 × TBS buffer (Bio-Rad, # 170-6435) containing 5% Blotting-Grade Blocker (Bio-Rad, # 170-6404) for 1 h. After washing with 1 × TBS buffer for 30 min, the membrane blot was incubated with the appropriately diluted primary antibody in antibody dilution buffer (1 × TBS, 5% BSA, 0.02% sodium azide) at 4 °C for 16 h. Then, the blot was washed three times with 1 × TBS (each time for 10 min) and incubated with secondary HRP-conjugated antibody in antibody dilution buffer (1:10,000 dilution) at room temperature for 1 h. After three washes with 1 × TBS (each time for 10 min), the blot was incubated with Clarity Western ECL Substrate (Bio-Rad, # 170-5060) for 1–2 min. The membrane was removed from the substrates and then exposed to the Amersham imager 600 (GE Healthcare Life Sciences, Marlborough, MA).

For immunoprecipitation, 2% of cell lysates were saved as an input control, and the remainder was incubated with 5–10 μl of the indicated antibody plus 20 μl of Pierce Protein A/G Plus Agarose (Thermo Fisher Scientific, # 20423) or 10 μl of EZview Red Anti-FLAG M2 Affinity Gel (Sigma, # F2426). After mixing end-over-end at 4 °C overnight, the beads were washed 3 times (5 min each wash) with 500 μl of lysis buffer. For ubiquitin detection, all beads were washed with 1 M urea for 15 min, 3 times to exclude potential binding of unanchored polyubiquitin.

**Protein purification from E. coli and pull-down assays.** The indicated FIP200 and RIG-I domains were cloned into pGEX-5X-3 (GE Healthcare, # 28-9545-55) to fuse with a GST tag, pET28b(+) (Novagen, # 69865-3) to fuse with a His tag, pT7-FLAG-2 (Sigma, # P1243) for a FLAG tag, or pMXB10 (New England Biolabs, # E6901S) for a MBP tag. These constructs were transfected into BL21 (DE3) E. coli (New England Biolabs, # C2527I) and cultured in LB broth at 20 °C. IPTG (0.4 mM) was added to induce protein expression. MBP-tagged proteins were purified using the IMPACT kit (New England Biolabs, # E6901S), and the MBP pull-down assays were performed using the anti-MBP Magnetic Beads New England Biolabs, # E8037S. FLAG-tagged proteins were purified by using the EZview Red Anti-FLAG M2 Affinity Gel (Sigma, # F2426). The GST Protein Interaction Pull-Down Kit (ThermoFisher Scientific, # PI21516) was used for GST-tagged protein purification and GST pull-down assays. The His-Spin Protein Miniprep kit (Zymo Research, # P2002) was used for His-tagged protein purification and His pull-down assays.

**Proteomics analysis of FIP200 protein complex.** AP-MS experiments were performed as previously described[49]. For protein purification, HEK293 cells stably expressing FLAG-tagged FIP200 were collected and lysed in 10 ml of the TAP lysis buffer. Cell lysates were pre-cleared with 50 μl of protein A/G resin before the addition of 20 μl of anti-FLAG resin (Sigma, # F2426) and incubated for 16 h at 4 °C on a rotator. The resin was washed three times and transferred to a spin column with 40 μl of the FLAG peptide for 1 h at 4 °C.

The purified samples were sent for mass spectrometry analysis. Proteins found in the control group were considered as non-specific binding proteins. The SAINT algorithm (http://sourceforge.net/projects/saint-apms) was used to evaluate the MS data. Proteins with SAINT score < 0.89 or with < 3 peptide hits are considered as non-specific binding proteins.

**Immunofluorescence assay.** Cells were cultured in the Lab-Tek II CC2 Chamber Slide System 4-well (Thermo Fisher Scientific, # 154917). After the indicated treatment, the cells were fixed and permeabilized in cold methanol for 10 min at −20 °C. Then, the slides were washed with 1 × PBS for 10 min and blocked with Odyssey Blocking Buffer (LI-COR Biosciences, # 927-40000) for 1 h. The slides were incubated in Odyssey Blocking Buffer with appropriately diluted primary antibodies at 4 °C for 16 h. After 3 washes (10 min per wash) with 1 × PBS, the cells were incubated with the corresponding Alexa Fluor conjugated secondary antibodies (Life Technologies) for 1 h at room temperature. The slides were washed three times (10 min each time) with 1 × PBS and counterstained with 300 nM DAPI for 1 min, followed by washing with 1 × PBS for 1 min. After air-drying, the slides were sealed with Gold Seal Cover Glass (Electron Microscopy Sciences, # 3223) using Fluoro-gel (Electron Microscopy Sciences, # 17985-10). Images were captured and analyzed using an iRiS™ Digital Cell Imaging System (Logos Biosystems).

**Proximity ligation assay.** The Proximity Ligation Assay (PLA) was performed using Duolink® In Situ Red Starter Kit Mouse/Rabbit (Sigma, # DUO92101-1KT) and Duolink® In Situ Detection Reagents Green (Sigma, # DUO92014) according to the manufacturer's protocol.

**Real-time PCR.** Total RNA was prepared using the RNeasy Mini Kit (Qiagen, # 74106). One μg quantity of RNA was reverse transcribed into cDNA using the QuantiTect reverse transcription kit (Qiagen, # 205311). For one real-time reaction, 10 μl of SYBR Green PCR reaction mix (Eurogentec) including 2 μg of the synthesized cDNA plus an appropriate oligonucleotide primer pair were analyzed on a 7500 Fast Real-time PCR System (Applied Biosystems). The comparative $Ct$ method was used to determine the relative mRNA expression of genes normalized by the housekeeping gene *GAPDH*. The primer sequences: human *GAPDH*, forward primer 5′- AGGTGAAGGTCGGAGTCA-3′, reverse primer 5′-GGTCATT GATGGCAACAA-3′; human *IFNb1*, forward primer 5′-TCATCCTGTCCTTGAG GCAGT-3′, reverse primer 5′-CAGCAATTTTCAGTGTCAGAAGC-3′; human *CXCL10 (IP10)*, forward primer 5′-TTCAAGGAGTACCTCTCTCTAG-3′, reverse primer 5′- CTGGATTCAGACATCTCTTCTC-3′; human *CCL5 (RANTES)* qPCR primers were purchased from Qiagen (# PPH00703B-200); mouse *Gapdh*, forward primer 5′-GCGGCACGTCAGATCCA-3′, reverse primer 5′- CATGGCCTTCCG TGTTCCTA-3′; mouse *Ifnb1*, forward primer 5′-CAGCTCCAAGAAAGGACG AAC-3′, reverse primer 5′-GGCAGTGTAACTCTTCTGCAT-3′; mouse *Cxcl10 (IP10)*, forward primer 5′- CCAAGTGCTGCCGTCATTTTC-3′, reverse primer 5′- GGCTCGCAGGGATGATTTCAA-3′; mouse *Ccl5 (RANTES)*, forward primer 5′- GCTGCTTTGCCTACCTCTCC-3′, reverse primer 5′-TCGAGTGACAAACAC GACTGC-3′; VSV, forward primer 5′-TGATACAGTACAATTATTTTGGGAC-3′, reverse primer 5′-GAGACTTTCTGTTACGGGATCTGG-3′, HCoV OC43, forward primer 5′- ATGTTAGGCCGATAATTGAGGACTAT-3′, reverse primer 5′-AATGTAAAGATGGCCGCGTATT-3′.

**RNA sequencing and data analysis**. Total RNA was prepared using the RNeasy Mini Kit (Qiagen, # 74106). RNA samples were sent to Novogene (Sacramento, CA) for sequencing. Data were analyzed in house as previously described[50]. Each sample was sequenced to generate a minimum of 20 million reads. The paired-end reads were directionally mapped to the mouse genome (GRCm38.p6) using TopHat2. Cufflink and CuffDiff analyses were performed to identify the differentially expressed genes using a fold change of ≥ 2 and an FDR of < 0.05. The RNA sequencing datasets were deposited to GEO (https://www.ncbi.nlm.nih.gov/geo) (Access number: GSE147001).

**ELISA**. The human IFNβ ELISA kit (PBL Assay Science, # 41410-2) and mouse IFNβ ELISA kit (PBL Assay Science, # 42400-2) were used. The ELISA assays were performed according to the manufacturer's protocols.

**Plasmid transfection**. HEK293 and A549 cells were transfected using Lipofectamine 3000 or Lipofectamine LTX Transfection Reagent (Life Technologies, # L3000015) according to the manufacturer's protocol.

**CRISPR/Cas9**. The single guide RNA (sgRNA) targeting sequences: human *FIP200* sgRNA: 5′-TTTCTAACAGCTCTATTACG-3′; mouse *Fip200* sgRNA: 5′-GTCAAATGTCAGCGTGGTTC -3′. The sgRNA was cloned into lentiCRISPR v2 vector[51] (Addgene). The lentiviral construct was transfected with psPAX2 and pMD2G into HEK293T cells using PEI. After 48 h, the media containing lentivirus were collected. The targeted cells were infected with the media containing the lentivirus supplemented with 10 µg ml$^{-1}$ polybrene. Cells were selected with 10 µg ml$^{-1}$ puromycin for 14 days. Single clones were expanded for knockout confirmation by Western blotting.

**Statistics and reproducibility**. The sample size was sufficient for data analyses using paired two-tailed Student's *t* test. For all statistical analysis, differences were considered to be statistically significant at values of $P < 0.05$.

**Reporting summary**. Further information on research design is available in the Nature Research Reporting Summary linked to this article.

## Data availability
All relevant data are within the manuscript and its supplementary files. Source data for figures are available in Supplementary Data 2. Uncropped western blots are provided in Supplementary Figs. 8–10. The RNA-seq data that support the findings of this study have been deposited in GEO with the accession number, GSE147001.

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

## Acknowledgements

The authors thank Dr. Herbert W. Virgin for the conditional knockout mouse strain, Drs. Neal DeLuca, Sean Whelan, Adolfo Garcia-Sastre, and David L. Bartlett for viruses, Drs. Noboru Mizushima and Do-Hyung Kim for constructs. This work was supported by the National Institutes of Health grants R01AI141399 (S.L. and J.G.), R15AI126360 (S. L.), R21AI137750 (S.L. and C.J.), P20GM103648 (L.L. and S.L.).

## Author contributions

S.L. conceived and supervised the project. S.L., L.W., K.S., W.H., and Y.W. designed the study. S.L., L.W., K.S., W.H., Y.W., G.P., C.W., L.L., and J.G. analyzed the data. C.H. and L.L. assisted analyzed the RNA Seq. J.R. assisted and examined the H&E slides. C.J. consulted and assisted mouse viral infection. J.G. provided the *Fip200^f/f* mice and consulted the project. All authors contributed to manuscript writing, revision, read, and approved the submitted version.

## Competing interests

The authors declare no competing interests.
