## [Peer Review File · Communications Biology]

Reviewers' comments:

Reviewer #1 (Remarks to the Author):

Comment to the authors

In this manuscript, the authors show that the cytosolic RNA sensor Retinoic acid-inducible gene I (RIG-I) is positively regulated by the FAK family kinase-interacting protein of 200 kDa (FIP200). The study demonstrates that FIP200, facilitates RIG-I activation following recognition of RNA molecules and production of a range of cytokines, including antiviral type I interferons. The authors provide very convincing evidence, supported by genetic knockout systems and in-vivo studies. This study provides insight into the novel regulatory mechanisms that can be exploited for the development of potential antiviral therapeutics.

Overall, based on conceptual advance and technical soundness, I find this manuscript a strong contribution to the field of innate immunity and infection biology. The following suggestions might further strengthen the conclusions of this study.

Major comments:

1. The authors provide substantial evidence that in response to poly(I:C), FIP200 promotes activation of RIG-I, however, poly(I:C) can trigger the activation of other RNA sensors such as Toll-like receptor 3 and Melanoma differentiation- associate gene 5 (Chow et al., *Annu Rev Immunol* 2018;36:667-694). Although the authors provided evidence of direct interaction between RIG-I and FIP200, it would be more relevant to provide genetic evidence looking at the specificity of FIP200 in promoting RIG-I activation or if it can modulate the outcome of other RNA sensing pathways. This can be tested in cells overexpressing FIP200 and deficient in RNA sensors (such as MDA5) or FIP200 KO cells complemented with FIP200 and deficient in RNA sensors (such as MDA5) and assaying the production of antiviral cytokines such as type I interferons.
2. In Figure 1c, the authors show quantification of in-situ interaction between RIG-I and FIP200 using proximity ligation assay (PLA). However, the images from this PLA experiment are not provided. Given that both RIG-I and FIP200 are distributed in various cytoplasmic locations, such as mitochondria and isolation membrane, respectively (Hara et al., *J Cell Biol.* 2008; 181(3): 497-510, Sánchez-Aparicio et al., *J Virol.* 2017; 91(2): e01155-16), it would be interesting to show the precise subcellular location of RIG-I and FIP200 interaction. This may be addressed by providing high resolution images from the PLA experiment.
3. In Figure 3m, the authors show that in response to poly(I:C) treatment, phosphorylation of TBK1 decreases in HEK293 cells over time. Although phosphorylation of TBK1 is decreased in FIP200 KO-1 cells, this decrease is not time-dependent, suggesting possible involvement of another signalling molecule controlling TBK1 phosphorylation. The authors should explain these differences and include densitometry quantification of the blots to provide quantitative evidence.
4. The statistical analysis is mostly sound and provided in the figures. However, some panels do not have information about the exact statistical tests used. For example, Figure 1c. The authors should provide appropriate details of the statistical tests used in each figure legend.

Minor comments:

1. The authors should use arrows/arrowheads to indicate the structural changes observed in supplementary Figure 6c. It would be clearer if the damage observed in the histology can be quantified.
2. The information about which protein is tagged with HA should be included in the legend of supplementary Figure 8a.
3. I suggest rerunning the top blot (IP: anti-FLAG WB: anti-MYC blot) in Figure 8a as the contrast of this blot is masking the actual background.
4. Line 197, the authors should briefly state the significance of using the d109 mutant of human simplex virus type 1.
5. Line 247, the sentence, "Next, we.... reconstituted...cells" should be revised to include "with" after the word "reconstituted".

Reviewer #2 (Remarks to the Author):

In this manuscript, Wang et al. found FIP200 interacted with the 2CARD of RIG-I. FIP200 facilitates RIG-I 2CARD release and dimerization, promoting RIG-I activation and host defense to RNA virus infection. Deficiency of FIP200 impaired dsRNA-induced type I IFN expression, and In vivo studies demonstrated FIP200 knockout mice were more susceptible to VSV and HCoV infection. The findings are interesting and the experiments are technically sound. However, this manuscript needs to elucidate following concerns in order to get published in Communication biology.

1. Fig1a and Fig1b showed that the interaction between endogenous FIP200 and RIG-I was weak in the absence poly(I:C) stimulus, but the IP-MS results found that FIP200 as a "high-confidence candidate interacting proteins" of RIG-I. In IP-MS experiment, no additional stimulus was added, the author needs to explain this contradiction. In addition, the mechanism explained by the authors suggests that the 2 cards of RIG-I should be in a self-inhibiting state when it is not activated, how does FIP200 interact with the card domain of RIG-I?
2. In Fig2, to demonstrate the role of FIP200 in RIG-I activation, the authors should include data showing RIG-I activators such poly(I:C), 5'-triphosphate dsRNA hairpin, SeV and IAV ΔNS, as well as negative controls such as ctDNA, poly(dG:dC), DMXAA, and LPS. Moreover, the authors should provide changes in downstream protein levels, such as pTBK1, pIRF3, etc. Personally, I think part 205-215 should be combined with this part (147-161).
3. In Fig3 and Fig4, the authors should provide data showing the cell viability. Complete knockout of FIP200 in mouse is embryonic lethal, I wonder if knocking out FIP200 in cells affects its viability or proliferation, etc.
4. In Fig6a, line 223 the authors have mentioned "ULK1 failed to activate RIG-I", but the infection activity of overexpressed ULK1 cells showed obvious difference from that of empty vector.
5. Fig4 and Fig5, Lack of evidence of FIP200 knockout/reconstitution efficiency.
6. Deletion of CC domain promotes CARD-CARD interaction(fig8e), why delCC had no effect on IFNB activation? (fig8f).
7. Fig3b-d, need clarify the meaning of the abscissa.

Reviewer #3 (Remarks to the Author):

This study reported that FIP200 plays a supportive role in the cytosolic RNA sensing pathway. The authors originally found that FIP200 may interact with RIG-I by AP-MS. Then they demonstrated with a series of genetic experiments that FIP200 plays a positive role in RNA sensing. The interaction was further characterized with biochemical studies, which lead to the proposed model that FIP200

promotes RIG-I oligomerization. The work is novel and should appeal to the general audience of the journal.

Comments:

- 1) Figure 8. The evidence that FIP200 promotes RIG-I oligomerization is weak. IP experiments show that overexpressed FIP200 can block the interaction between 2CARD and Hel-CTD, and can promote 2CARD dimerization. The data is insufficient to support the claim on FIP200's role in RIG-I oligomerization. The authors should demonstrate the effect with purified RIG-I and FIP200 in assays like gel filtration or analytical ultracentrifugation.
- 2) Figure 1b. Please show separate channels of microscopy images.
- 3) Figure 1c. Please show representative microscopy images of the PLA assay.

Communications Biology
COMMSBIO-21-0412

Title: FIP200 restricts RNA virus infection by facilitating RIG-I activation

Responses to the Reviewers comments (comments from reviewers in black, responses to the reviewers in blue, changes in the text are marked in blue)

We thank the reviewers for the constructive comments and suggestions to improve our manuscript. We agreed to most of the suggestions and comments. We have carried out additional experiments and made modifications in the text to improve the manuscript quality. Our point-by-point responses are included as the following.

Reviewer comments:

Reviewer #1 (Remarks to the Author):

Comment to the authors

In this manuscript, the authors show that the cytosolic RNA sensor Retinoic acid-inducible gene I (RIG-I) is positively regulated by the FAK family kinase-interacting protein of 200 kDa (FIP200). The study demonstrates that FIP200, facilitates RIG-I activation following recognition of RNA molecules and production of a range of cytokines, including antiviral type I interferons. The authors provide very convincing evidence, supported by genetic knockout systems and in-vivo studies. This study provides insight into the novel regulatory mechanisms that can be exploited for the development of potential antiviral therapeutics.

Overall, based on conceptual advance and technical soundness, I find this manuscript a strong contribution to the field of innate immunity and infection biology. The following suggestions might further strengthen the conclusions of this study.

Major comments:

1. The authors provide substantial evidence that in response to poly(I:C), FIP200 promotes activation of RIG-I, however, poly(I:C) can trigger the activation of other RNA sensors such as Toll-like receptor 3 and Melanoma differentiation- associate gene 5 (Chow et al., Annu Rev Immunol 2018;36:667-694). Although the authors provided evidence of direct interaction between RIG-I and FIP200, it would be more relevant to provide genetic evidence looking at the specificity of FIP200 in promoting RIG-I activation or if it can modulate the outcome of other RNA sensing pathways. This can be tested in cells overexpressing FIP200 and deficient in RNA sensors (such as MDA5) or FIP200 KO cells complemented with FIP200 and deficient in RNA sensors (such as MDA5) and assaying the production of antiviral cytokines such as type I interferons.

We used low molecular weight (LMW) poly(I:C) and 5'-ppp-dsRNA as RIG-I ligands. Although poly(I:C) is a ligand for RIG-I and MDA5, LMW poly(I:C) is roughly recognized by RIG-I while high molecular weight (HMW) poly(I:C) is sensed by MDA5. The non-transfected poly(I:C) is a ligand for TLR3. Throughout the manuscript, we stimulated cells with transfected LMW poly(I:C). We now clarify it in the methods.

To answer the reviewer's question, we tested poly(A:U), an exclusive ligand for TLR3 in FIP200 WT and KO macrophages. We found that FIP200 deficiency had little effect on TLR3 (Supplemental Fig. 3d). We also tested HMW poly(I:C) by transfection. Interestingly, FIP200

knockout also impaired MDA5-induced IFN β expression (Supplemental Fig. 3c). Because of the high similarity of CARD sequences of RIG-I and MDA5, we suspected that MDA5 also interacted with FIP200. Co-IP confirmed that FIP200 also interacted with the 2CARD of MDA5 (Supplemental Fig. 1f). Future work will investigate the role of FIP200 in MDA5 signaling.

2. In Figure 1c, the authors show quantification of in-situ interaction between RIG-I and FIP200 using proximity ligation assay (PLA). However, the images from this PLA experiment are not provided. Given that both RIG-I and FIP200 are distributed in various cytoplasmic locations, such as mitochondria and isolation membrane, respectively (Hara et al., J Cell Biol. 2008; 181(3): 497–510, Sánchez-Aparicio et al., J Virol. 2017; 91(2): e01155-16), it would be interesting to show the precise subcellular location of RIG-I and FIP200 interaction. This may be addressed by providing high resolution images from the PLA experiment.

We repeated the PLA with mitochondria tracker to indicate the mitochondria (Supplemental Fig. 1e). We observed more PLA signals (green dots) in the mitochondria after poly(I:C) stimulation.

3. In Figure 3m, the authors show that in response to poly(I:C) treatment, phosphorylation of TBK1 decreases in HEK293 cells over time. Although phosphorylation of TBK1 is decreased in FIP200 KO-1 cells, this decrease is not time-dependent, suggesting possible involvement of another signalling molecule controlling TBK1 phosphorylation. The authors should explain these differences and include densitometry quantification of the blots to provide quantitative evidence. Figure 3m examined whether poly(I:C) induced TBK1 phosphorylation in FIP200 KO cells. We analyzed the band densitometry by Image J and calculated the ratio of p-TBK1 to total TBK1 (Fig. 3m). The result showed that TBK1 was barely phosphorylated in FIP200 KO cells, and there might be a little bit increase at 2 h. In the FIP200 WT cells, TBK1 phosphorylation was strongly induced and time-dependent (Fig. 3m).

4. The statistical analysis is mostly sound and provided in the figures. However, some panels do not have information about the exact statistical tests used. For example, Figure 1c. The authors should provide appropriate details of the statistical tests used in each figure legend.

Added.

Minor comments:

1. The authors should use arrows/arrowheads to indicate the structural changes observed in supplementary Figure 6c. It would be clearer if the damage observed in the histology can be quantified.

Arrows added. The histology in *Fip200^{f/f};LysM-Cre* mouse is dramatically different from control mouse. We added a more detailed description in the text: “*Fip200^{f/f};LysM-Cre* mouse exhibited a marked diffuse interstitial pneumonia with focal areas of vasculitis and bronchitis with mixed polynuclear/mononuclear infiltrates and scattered areas of epithelial necrosis whereas lung tissue obtained from *Fip200^{f/f}* mouse was essentially normal”.

2. The information about which protein is tagged with HA should be included in the legend of supplementary Figure 8a.

Clarified.

3. I suggest rerunning the top blot (IP: anti-FLAG WB: anti-MYC blot) in Figure 8a as the contrast of this blot is masking the actual background.

Replaced with a new one.

4. Line 197, the authors should briefly state the significance of using the d109 mutant of human simplex virus type 1.

Added.

5. Line 247, the sentence, "Next, we.... reconstituted...cells" should be revised to include "with" after the word "reconstituted".

Added.

Reviewer #2 (Remarks to the Author):

In this manuscript, Wang et al. found FIP200 interacted with the 2CARD of RIG-I. FIP200 facilitates RIG-I 2CARD release and dimerization, promoting RIG-I activation and host defense to RNA virus infection. Deficiency of FIP200 impaired dsRNA-induced type I IFN expression, and In vivo studies demonstrated FIP200 knockout mice were more susceptible to VSV and HCoV infection. The findings are interesting and the experiments are technically sound. However, this manuscript needs to elucidate following concerns in order to get published in Communication biology.

1. Fig1a and Fig1b showed that the interaction between endogenous FIP200 and RIG-I was weak in the absence poly(I:C) stimulus, but the IP-MS results found that FIP200 as a "high-confidence candidate interacting proteins" of RIG-I. In IP-MS experiment, no additional stimulus was added, the author needs to explain this contradiction. In addition, the mechanism explained by the authors suggests that the 2 cards of RIG-I should be in a self-inhibiting state when it is not activated, how does FIP200 interact with the card domain of RIG-I?

We observed an interaction, although weakly, between RIG-I and FIP200 without poly(I:C) stimulation. In the IP-MS experiment, high-confidence candidate interacting proteins are defined by SAINT analysis. The confidence is built on the SAINT score that compares all protein complexes in the database by integrating several parameters, such as occurrence, reproducibility, spectral counts, etc. It is not necessary that a high-confidence interaction must be a strong interaction.

Several structural studies suggest that the second CARD domain binds Hel-CTD to keep RIG-I inert (we added a sentence for clarification. Line 288). First, the first CARD could interact with FIP200. Second, like the protein-protein interaction, the CARD binding to Hel-CTD is not static and determined by the dissociation constant (Kd). Third, FIP200 could bind the sites in the second CARD, which are not occupied by Hel-CTD. FIP200 also can compete with Hel-CTD for binding the same sites.

2. In Fig2, to demonstrate the role of FIP200 in RIG-I activation, the authors should include data showing RIG-I activators such poly(I:C), 5'-triphosphate dsRNA hairpin, SeV and IAV ΔNS, as well as negative controls such as ctDNA, poly(dG:dC), DMXAA, and LPS. Moreover, the authors should provide changes in downstream protein levels, such as pTBK1, pIRF3, etc. Personally, I think part 205-215 should be combined with this part (147-161).

Fig.2 shows the synergy between FIP200 and RIG-I under overexpression conditions. When RIG-I is overexpressed, it is activated even though we keep the activation magnitude at minimal levels. We added two figure panels (Figs. 2c and 2f) to compare RIG-I-induced pTBK1 levels with vs. without FIP200.

3. In Fig3 and Fig4, the authors should provide data showing the cell viability. Complete knockout of

FIP200 in mouse is embryonic lethal, I wonder if knocking out FIP200 in cells affects its viability or proliferation, etc.

MTT assays are added to show the comparable growth rate between FIP200 wild type and knockout cells (Supplemental Figs. 2b, 2g, 3b).

4. In Fig6a, line 223 the authors have mentioned “ULK1 failed to activate RIG-I”, but the infection activity of overexpressed ULK1 cells showed obvious difference from that of empty vector.

We agree with the reviewer that there is a difference between vector/RIG-I and ULK1/RIG-I in Fig. 6a. The data showed that ULK1 could not activate RIG-I, instead, might inhibit RIG-I, thereby increasing VSV infection. How ULK1 inhibits RIG-I is beyond the topic of this manuscript and will be investigated in the future.

5. Fig4 and Fig5, Lack of evidence of FIP200 knockout/reconstitution efficiency.

Western blots are added to show the efficiency (Figs. 5b, 5d, Supplemental Figs. 2a, 2h, 3a, 7h)

6. Deletion of CC domain promotes CARD-CARD interaction(fig8e), why delCC had no effect on IFN β activation? (fig8f).

The old Fig. 8f showed that full-length FIP200, but not delCC, strongly promoted the CARD-CARD interaction. We repeated the IP experiment and increased the washing strength. The new data showed that delCC failed to promote CARD-CARD interaction.

7. Fig3b-d, need clarify the meaning of the abscissa.

Clarified.

Reviewer #3 (Remarks to the Author):

This study reported that FIP200 plays a supportive role in the cytosolic RNA sensing pathway. The authors originally found that FIP200 may interact with RIG-I by AP-MS. Then they demonstrated with a series of genetic experiments that FIP200 plays a positive role in RNA sensing. The interaction was further characterized with biochemical studies, which lead to the proposed model that FIP200 promotes RIG-I oligomerization. The work is novel and should appeal to the general audience of the journal.

Comments:

1) Figure 8. The evidence that FIP200 promotes RIG-I oligomerization is weak. IP experiments show that overexpressed FIP200 can block the interaction between 2CARD and Hel-CTD, and can promote 2CARD dimerization. The data is insufficient to support the claim on FIP200's role in RIG-I oligomerization. The authors should demonstrate the effect with purified RIG-I and FIP200 in assays like gel filtration or analytical ultracentrifugation.

We have performed sucrose-gradient centrifugation to examine the oligomerized forms of RIG-I. As shown in Supplemental Fig. 7d, poly(I:C) induced RIG-I to shift to the high molecular weight fractions in wild type cells, but not in FIP200 knockout cells.

2) Figure 1b. Please show separate channels of microscopy images.

Added.

3) Figure 1c. Please show representative microscopy images of the PLA assay.

Added (in Supplemental Fig. 1e).

REVIEWERS' COMMENTS:

Reviewer #1 (Remarks to the Author):

The authors have carried out additional experiment and have addressed all the comments. I have no further comments on the revised version.

Reviewer #2 (Remarks to the Author):

The authors have addressed all my concerns. One minor point: the IP exp in Fig1e, there is a lack of a Lysate WB: anti-Flag blot; also in the 1st "IP: anti-Flag WB: anti-HA" blot, the authors should show the >75kd part, the Hel-CTD co-IP info is missing in the current blot.

Reviewer #3 (Remarks to the Author):

The authors have address my previous concerns.

Communications Biology
COMMSBIO-21-0412

Title: FIP200 restricts RNA virus infection by facilitating RIG-I activation

Responses to the Reviewers comments

Reviewer comments:

Reviewer #1 (Remarks to the Author):

The authors have carried out additional experiment and have addressed all the comments. I have no further comments on the revised version.

Reviewer #2 (Remarks to the Author):

The authors have addressed all my concerns. One minor point: the IP exp in Fig1e, there is a lack of a Lysate WB: anti-Flag blot; also in the 1st "IP: anti-Flag WB: anti-HA" blot, the authors should show the >75kd part, the Hel-CTD co-IP info is missing in the current blot.

We now add the anti-FLAG lysate blot and also show the >75 kD part of the IP blot in Fig. 1e.

Reviewer #3 (Remarks to the Author):

The authors have address my previous concerns.